# Beyond Primal-Dual Methods in Bandits with Stochastic and Adversarial Constraints

**Martino Bernasconi**[†] **Matteo Castiglioni**[‡] **Andrea Celli**[†] **Federico Fusco**[*]

[†] Bocconi university
[‡] Politecnico di Milano
[*] Sapienza University of Rome

{martino.bernasconi,andrea.celli2}@unibocconi.it,
matteo.castiglioni@polimi.it, federico.fusco@uniroma1.it

## Abstract

We address a generalization of the bandit with knapsacks problem, where a learner aims to maximize rewards while satisfying an arbitrary set of long-term constraints. Our goal is to design best-of-both-worlds algorithms that perform optimally under both stochastic and adversarial constraints. Previous works address this problem via primal-dual methods, and require some stringent assumptions, namely the Slater's condition, and in adversarial settings, they either assume knowledge of a lower bound on the Slater's parameter, or impose strong requirements on the primal and dual regret minimizers such as requiring weak adaptivity. We propose an alternative and more natural approach based on optimistic estimations of the constraints. Surprisingly, we show that estimating the constraints with an UCB-like approach guarantees optimal performances. Our algorithm consists of two main components: (i) a regret minimizer working on *moving strategy sets* and (ii) an estimate of the feasible set as an optimistic weighted empirical mean of previous samples. The key challenge in this approach is designing adaptive weights that meet the different requirements for stochastic and adversarial constraints. Our algorithm is significantly simpler than previous approaches, and has a cleaner analysis. Moreover, ours is the first best-of-both-worlds algorithm providing bounds logarithmic in the number of constraints. Additionally, in stochastic settings, it provides $\widetilde{O}(\sqrt{T})$ regret *without* Slater's condition.

## 1 Introduction

We address the problem faced by a decision maker who aims to maximize its cumulative reward over a time horizon $T$, while satisfying an arbitrary set of $m$ long-term constraints. At each round $t$, the learner selects an action $a_t$ from a finite set of $K$ actions, and then observes a reward $f_t(a_t)$ and some costs $g_t(a_t) \in [-1, 1]^m$. The goal is to design best-of-both-worlds algorithms for this problem that perform optimally under both stochastic and adversarial constraints. We always assume rewards are generated adversarially. This is because the real complexity of the problem is captured by the nature of the constraints, so that transitioning from adversarial to stochastic rewards under the same type of constraints does not affect our results.

The first works on bandits with constraints focus on budget constraints, a.k.a bandit with knapsack (BwK) [7] study the settings in which both rewards and constraints are i.i.d. and propose an UCB-based approach, combined with primal-dual method. Agrawal and Devanur [2] provide an UCB-like approach for more general rewards and costs. Immorlica et al. [21], Kesselheim and Singla [22]

analyse settings with adversarial constraints and rewards, providing a primal-dual algorithm to tackle the problem. Castiglioni et al. [15] show that a similar primal-dual approach provides best-of-both-worlds guarantees. Many subsequent works extend the setting to more general constraints, mostly employing primal-dual methods [16, 17, 28, 11, 13, 10, 18]. Primal-dual methods have been the only effective method that provides best-of-both-worlds guarantees for bandits with constraints [16, 17, 11, 13, 10]. However, such methods require assumptions that are particularly stringent in settings beyond knapsack constraints. First, they require the existence of a strictly feasible solution (*i.e.,* Slater's condition) to avoid a regret of order $O(T^{3/4})$ [16, 28]. While this assumption always holds in bandits with knapsack setting (where "doing nothing" incurs in a negative cost equal to the per-round budget), this assumption is far more stringent with general constraints. Moreover, some works require the knowledge of a lower bound on the Slater's parameter [16, 28]. Subsequent works circumvent this assumption at the expense of strong requirements on the primal and dual regret minimizers [17, 11, 13, 1]. In particular, such approaches require weakly-adaptive primal and dual regret minimizers. The challenge of applying such primal-dual algorithms to bandit beyond knapsack constraint is reflected in regret bounds that exhibit non-optimal dependencies on some parameters. For instance, a polynomial (instead of logarithmic) dependence on the number of constraints [17, 11, 13]. For further pointers to the literature, we refer to Appendix A.

## 1.1 Our contribution

We propose an alternative and insightful approach to design best-of-both-worlds algorithms for bandit with long-term constraints. Our method relies on optimistic estimations of the constraints through a weighted empirical mean of past samples. Surprisingly, we demonstrate that using a UCB-based approach to estimate the constraints ensures optimal performance under both stochastic and adversarial constraints. Our algorithm differs significantly from previous UCB-based approaches. For instance, it guarantees no-regret even with adversarial rewards and stochastic constraints, unlike previous works [2, 7, 23]. Moreover, it is the first UCB-like approach that provides an optimal competitive ratio of $1 + 1/\rho$ with adversarial constraints, where $\rho$ is the unknown Slater's parameter.

Our algorithm consists of two simple components. The first is an adversarial regret minimizer working on *moving strategy sets*. In particular, at each round, the regret minimizer chooses a strategy in the current optimistic estimation of the feasible set, and is required to achieve no-regret with respect to any strategy in the intersection of all feasibility set estimations. The second component is a tool for estimating the feasible set using an optimistic weighted mean of previous samples. The key challenge in this approach is designing adaptive weights that meet the different requirements for stochastic and adversarial constraints. Intuitively, in stochastic settings, we aim to converge to the (unweighted) empirical mean of the observed constraints. Conversely, in adversarial settings, we should assign larger weights to recent samples to address time-dependent constraints.

Not only is our algorithm significantly simpler than previous approaches, with a clean and insightful analysis, but it also provides better theoretical performance than primal-dual methods. Indeed, it is the first best-of-both-worlds algorithm to provide bounds logarithmic in the number of constraints. Moreover, in stochastic settings, it is the first algorithm to provide $\widetilde{O}(\sqrt{T})$ regret *without* requiring Slater's condition. Finally, it guarantees that the expected violation in the current round converges to zero, making our algorithm "converge" to strategies that are feasible in expectation. This provides a more stable and consistent control on the violations.

## 2 Model and Preliminaries

We address the problem faced by an agent aiming at maximizing its cumulative reward over a time horizon $T$, while satisfying $[\![m]\!]$ long-term constraints.[1] The agent has a set $[\![K]\!]$ of available actions and, at each round $t \in [\![T]\!]$, selects $a_t \in [\![K]\!]$. The agent then observes the corresponding reward $f_t(a_t) \in [0, 1]$ and a cost $g_t^{(i)}(a_t) \in [-1, 1]$, for each constraint $i \in [\![m]\!]$. We define the cumulative violation of the $i^{th}$ constraint as

$$V_T^{(i)} := \sum_{t \in [\![T]\!]} g_t^{(i)}(a_t),$$

---

[1] For any $N \in \mathbb{N}$, we use $[\![N]\!]$ to denote the set $\{1, \dots, N\}$.

while $V_T := \max_{i\in[\![m]\!]} V_T^{(i)}$ is the maximum violation across all constraints. At a high level, we want to minimize the regret while keeping the violation of each constraint $V_T^{(i)}$ sublinear in $T$.

The focus of this paper is on handling both stochastic and adversarial constraints. Conversely, we always assume the rewards to be generated up-front by an adversary; we do not treat explicitly the situation where the rewards are generated i.i.d. because our guarantees are already tight for the harder case of adversarial rewards.[2] In the stochastic setting, we assume that $g_t = \{g_t^{(i)}\}_{i\in[\![m]\!]}$ is drawn i.i.d. from a fixed but unknown distribution $\mathcal{G}$, and we let $\bar{g}^{(i)}(a) = \mathbb{E}_{g\sim\mathcal{G}}[g^{(i)}(a)]$ be the expected cost of action $a$ for the $i^{th}$ constraint. On the other hand, in the adversarial setting $\{g_t\}_{t\in[\![T]\!]}$ is an arbitrary sequence of cost functions.

Let $\Delta_K$ to be the set of discrete probability distributions over the set $[\![K]\!]$. Then, at round $t \in [\![T]\!]$, given a randomized strategy $x_t \in \Delta_K$, the expected learner reward is $\sum_{a\in[\![K]\!]} f_t(a)x_t(a) = \langle x_t, f_t \rangle$. Similarly, $\langle x_t, g_t^{(i)} \rangle$ denotes the expected cost of the $i^{th}$ constraint. Finally, we use $n_t(a)$ to denote the number of times arm $a$ was played up to time $t$, i.e., $n_t(a) = \sum_{\tau=1}^{t} \mathbb{I}(a_\tau = a)$.

We want to design algorithms which achieve good performances in both the adversarial and the stochastic setting. As it is customary in the literature, we compare our learning algorithm with different benchmarks according to the setting.

**Stochastic Benchmark** In the stochastic setting, the constraints $g_t^{(i)}$ are i.i.d. samples with mean $\bar{g}^{(i)}$ and thus we consider as benchmark the best fixed randomized strategy that satisfies the constraints in expectation, which is a standard choice in bandits with constraints [11, 28, 16]. Formally, in the stochastic setting, we can define the feasible sets $\mathcal{X}_i^\star$ and $\mathcal{X}^\star$ as follows:

$$\mathcal{X}_i^\star := \left\{ x \in \Delta_K : \langle x, \bar{g}^{(i)} \rangle \le 0 \right\} \quad \text{and} \quad \mathcal{X}^\star := \cap_{i\in[\![m]\!]} \mathcal{X}_i^\star.$$

Then, we can define the stochastic baseline as:

$$\mathsf{OPT_S} := \max_{x\in\mathcal{X}^\star} \sum_{t\in[\![T]\!]} \langle x, f_t \rangle.$$

We naturally assume the existence of safe mixed strategies, *i.e.,* that $\mathcal{X}^\star \ne \emptyset$. This is equivalent to assume the existence of a randomized strategy $x^\varnothing$ such that $\langle x^\varnothing, \bar{g}^{(i)} \rangle \le 0$ for all $i$. Notice that this is a weaker assumption than the one commonly assumed by best-of-both-worlds algorithms in which $\langle x^\varnothing, \bar{g}^{(i)} \rangle \le -\rho$, where $\rho$ is a strictly positive constant (see, e.g., [16, 11, 10]).

**Adversarial Benchmark** In the adversarial setting, $\{g_t\}_{t\in[\![T]\!]}$ is an arbitrary sequence of constraints. We consider as benchmark the best unconstrained strategy:

$$\mathsf{OPT_A} := \max_{x\in\Delta_K} \sum_{t\in[\![T]\!]} \langle x, f_t \rangle.$$

While this baseline has already been used [e.g., 11, 13]), other works on adversarial bandit with constraints employ weaker baselines [e.g., 21, 16]. For instance, Castiglioni et al. [16] consider the best fixed strategy which is feasible on average. However, we show that, despite using a stronger baseline, we obtain a competitive ratio that is optimal even for the weaker baselines commonly adopted in the literature [16, 10, 9].

## 2.1 Best-Of-Both-Worlds Guarantees

Our goal is to design learning algorithms that exhibit optimal guarantees both in the stochastic and adversarial settings. In the stochastic setting, we are interested in minimizing the *regret* $R_T$ w.r.t. $\mathsf{OPT_S}$:

$$R_T = \mathsf{OPT_S} - \sum_{t\in[\![T]\!]} f_t(a_t),$$

---

[2]Indeed, when the constraints are stochastic, we obtain the state-of-the-art $\widetilde{O}(\sqrt{T})$ regret even with adversarial rewards.

---

**Algorithm 1**

---

**Require:** bonuses $b_t(a)$, weights $w_{t,a}^{(i)}$ and parameter $\beta$
 1: Initialize regret minimizer $\mathcal{R}$ with $\beta$
 2: **for** each step $t = 1, \ldots, T$ **do**
 3:   **Estimation:**
 4:     $\hat{g}_t^{(i)}(a) \leftarrow \sum_{\tau \in \mathcal{T}_{t-1,a}} w_{t,a}^{(i)}(\tau) g_\tau^{(i)}(a)$ for all $a \in [\![K]\!]$ and $i \in [\![m]\!]$
 5:     $\widehat{\mathcal{X}}_t^{(i)} \leftarrow \{x \in \Delta_K : \langle x, \hat{g}_t^{(i)} - b_t \rangle \leq 0\}$
 6:     $\widehat{\mathcal{X}}_t \leftarrow \cap_{i \in [\![m]\!]} \widehat{\mathcal{X}}_t^{(i)}$
 7:   **Regret minimization:**
 8:     Get prediction from $\mathcal{R}$ on set $\widehat{\mathcal{X}}_t$: $x_t \leftarrow \mathcal{R}(\widehat{\mathcal{X}}_t)$
 9:   Sample $a_t \sim x_t$ and receive $\{g_t^{(i)}(a_t)\}_{i \in [\![m]\!]}$ and $f_t(a_t)$

---

and specifically we require both $R_T$ and $V_T$ to be in $\widetilde{O}(\sqrt{T})$ with high probability. This clearly matches the standard $\Omega(\sqrt{T})$ lower bound that holds even without constraints [5].

In the (harder) adversarial setting, we pose the less ambitious goal of achieving a constant competitive ratio with respect to $\mathsf{OPT}_\mathsf{A}$, or equivalently sublinear $\alpha$-regret with constant $\alpha$. Formally, given an $\alpha < 1$, we define the $\alpha$-regret as:

$$\alpha\text{-}R_T = \alpha \cdot \mathsf{OPT}_\mathsf{A} - \sum_{t \in [\![T]\!]} f_t(a_t).$$

As it is customary in the literature [15], the competitive ratio $\alpha$ obtained by our algorithms depends on the following Slater's parameter $\rho$:

$$\rho = - \inf_{a \in [\![K]\!]} \max_{t \in [\![T]\!], i \in [\![m]\!]} g_t^{(i)}(a). \tag{1}$$

The parameter $\rho$ is related to the existence of strictly-feasible actions, and only depends on the constraints. Our definition is slightly stronger than the one in most previous works where the $\inf$ is over randomized strategies. To guarantee the existence of a feasible strategy we assume that $\rho \geq 0$. Then, our goal is to guarantee that both $V_T$ and the $\alpha$-regret, with $\alpha = \rho/\rho+1$, belong to $\widetilde{O}(\sqrt{T})$ with high probability. Note that this matches the lower bound of Bernasconi et al. [11].

## 3   Our Approach

In this section, we present the main components of our algorithm, while the following sections will describe the specific components in details. We refer to Algorithm 1 for the pseudocode. At each step $t$, the algorithm works in two phases: i) it estimates the feasible set, and ii) it plays a strategy in the estimated set. Each phase requires a specific ingredient:

  i) An estimator $\hat{g}_t^{(i)}$ of the costs functions $g_t^{(i)}$ that is used together with the optimistic bonus $b_t$ to define the estimation of the feasible set defined as $\widehat{\mathcal{X}}_t := \cap_{i \in [\![m]\!]} \widehat{\mathcal{X}}_t^{(i)}$. In the stochastic case, we would like $\widehat{\mathcal{X}}_t \supseteq \mathcal{X}^\star$, while in the adversarial case our goal is to maintain a sequence of sets that always contains a version of the action set $\mathcal{X}$, properly scaled around $a^\varnothing$ (see Equation (2) for a formal definition).

  ii) A regret minimizer $\mathcal{R}$ for adversarial linear reward function that, at each round, takes in input a convex set of feasible strategies $\widehat{\mathcal{X}}_t \subseteq \Delta_K$, and then selects a strategy $x_t \in \widehat{\mathcal{X}}_t$. We require the regret minimizer to achieve $\widetilde{O}(\sqrt{KT})$ regret with respect to any $x \in \cap_{t \in [\![T]\!]} \widehat{\mathcal{X}}_t$;

In the following we define the two phases more in details. Let $\mathcal{T}_{t,a} := \{\tau \leq t : a_t = a\}$ be the set of rounds in which the algorithm plays action $a$. Then, at each round $t$, Algorithm 1 computes the estimate

$$\hat{g}_t^{(i)}(a) = \sum_{\tau \in \mathcal{T}_{t-1,a}} w_{t,a}^{(i)}(\tau) g_\tau(a) \quad \forall a \in [\![K]\!] \text{ and } i \in [\![m]\!]$$

---

**Algorithm 2** No Regret on Moving Sets

---

**Require:** Parameter $\beta > 0$
1: Set $\gamma = \beta/2$
2: **for** each step $t = 1, \ldots, T$ **do**
3:     Receive $\widehat{\mathcal{X}}_t$
4:     $\hat{x}_t(a) \leftarrow x_{t-1}(a) e^{\beta(\hat{f}_{t-1}(a)-1)}$ for all $a \in [\![K]\!]$
5:     $x_t \leftarrow \Pi_{\widehat{\mathcal{X}}_t}(\hat{x}_t) \coloneqq \arg\min_{x \in \widehat{\mathcal{X}}_t} B(x || \hat{x}_t)$
6:     Sample $a_t \sim x_t$
7:     Observe $f_t(a_t)$ and set $\hat{f}_t(a) \leftarrow 1$ for all $a \neq a_t$ and $\hat{f}_t(a_t) \leftarrow 1 - \frac{1 - f_t(a_t)}{x_t(a_t) + \gamma}$

---

as the weighted mean of *available* past observations $\{g_\tau^{(i)}(a)\}_{\tau \in [\![t-1]\!]}$ for each actions $a \in [\![K]\!]$ and constraint $i \in [\![m]\!]$, for some weights $w_{t,a}^{(i)}$.[3] Then, the estimates together with the optimistic bonus $\{b_t(a)\}_{a \in [\![K]\!]}$ are used to define the *moving sets* $\widehat{\mathcal{X}}_t$, which are fed to the regret minimizer $\mathcal{R}$ which in turn selects a point $x_t \in \widehat{\mathcal{X}}_t$.

One crucial property that is required for the execution of the regret minimizer $\mathcal{R}$ is that all the sets $\widehat{\mathcal{X}}_t$ are non-empty (as otherwise the regret minimizer has no feasible strategies). To simplify exposition, in the following sections we assume that the clean event $\mathcal{C} \coloneqq \left\{ \widehat{\mathcal{X}}_t \neq \{\emptyset\} \, \forall t \in [\![T]\!] \right\}$ holds. In Corollary 6.3, we prove that this event holds with high probability in the stochastic setting, while in Theorem 5.2 we argue that it holds deterministically in the adversarial one.

In Algorithm 1 we left unspecified two crucial parts of our approach. The first is how to build the regret minimizer $\mathcal{R}$, and the second concerns how to actually generate the sets $\widehat{\mathcal{X}}_t$, i.e., the weights $w_{t,a}^{(i)}$ and the bonus $b_t(a)$. We delve into these details in Section 4 and Section 5, respectively.

## 4   No-regret on moving sets

We describe the regret minimizer $\mathcal{R}$ that exhibits no-regret with respect to any $x \in \cap_{t \in [\![T]\!]} \widehat{\mathcal{X}}_t$. We achieve this via a simple modification to the EXP-IX algorithm of Neu [25] that provides high probability results for multi-armed bandits via implicit exploration. More specifically, our algorithm maintains a randomized strategy $x_t \in \Delta_K$ which is updated using the biased reward estimate $\hat{f}_t(a)$ as in Neu [25] and then projected onto $\widehat{X}_t$ according to the negative entropy Bregman divergence $B(x||y) = \sum_{a \in [\![K]\!]} [x(a) \log (x(a)/y(a)) - x(a) + y(a)]$. We refer to Algorithm 2 for the pseudocode, and present here the main result of the Section.

**Theorem 4.1.** *Let $x_t$ be selected accordingly to Algorithm 2 run with arbitrary sequence of convex sets $\widehat{\mathcal{X}}_t \subseteq \Delta_K$ with $\gamma = \frac{\beta}{2}$ and $\beta = \sqrt{\frac{\log(K/\delta_1)}{KT}}$. Then, with probability at least $1 - \delta_1$ it holds that*

$$\sum_{t \in [\![T]\!]} \langle f_t, x \rangle - f_t(a_t) \leq 4\sqrt{KT \log(K/\delta_1)}, \qquad \forall x \in \bigcap_{t \in [\![T]\!]} \widehat{\mathcal{X}}_t.$$

This result establishes no-regret in the case of moving sets, taking as benchmark the optimal strategy in the intersection of all sets. To exploit this result in Algorithm 1, we have to make sure that in both the stochastic and adversarial setting the intersection of the sets $\widehat{\mathcal{X}}_t$ contains "good" strategies. In the stochastic setting, we show that with high probability it includes $\mathcal{X}^\star$, while, in the adversarial setting, it includes a strategy with utility $\rho/1+\rho \cdot \mathsf{OPT}_A$.

## 5   How to build the sets $\widehat{\mathcal{X}}_t$

In this section, we show how to design estimations $\widehat{\mathcal{X}}_t$ of the feasible sets that, surprisingly, are effective both in stochastic and adversarial settings. Indeed, the main challenge is to design sets $\widehat{\mathcal{X}}_t$

---

[3]If a given action has been played at least once, we require $\sum_{\tau \in \mathcal{T}_{t-1,a}} w_{t,a}^{(i)}(\tau) = 1$, i.e., that $\hat{g}_t^{(i)}(a)$ is actually a weighted mean. Otherwise, the estimation is simply set to 0.

that accommodate the different requirements of the two settings. First, in Section 5.1, we discuss how to set the optimistic bonuses $b_t$ and then in Section 5.2 we focus on how to set the weights $w_{t,a}^{(i)}$.

## 5.1 How to set the optimistic bonus

The optimistic bonuses have the main purpose of balancing the estimation error in the stochastic setting. As the following lemma show, we simply need that $|\hat{g}_t^{(i)}(a) - \bar{g}^{(i)}(a)| \leq b_t(a)$ with high probability. Indeed, this is sufficient to show that $\mathcal{X}^\star \subseteq \cap_{t \in [\![T]\!]} \widehat{\mathcal{X}}_t$ in the stochastic setting.

**Theorem 5.1.** *Consider the stochastic setting. Given any $\delta > 0$, let $b_t(a)$ be such that with probability at least $1 - \delta$ it holds:*

$$|\hat{g}_t^{(i)}(a) - \bar{g}^{(i)}(a)| \leq b_t(a) \quad \forall t \in [\![T]\!], i \in [\![m]\!], a \in [\![K]\!].$$

*Then, it holds $\mathcal{X}^\star \subseteq \cap_{t \in [\![T]\!]} \widehat{\mathcal{X}}_t$ with probability at least $1 - \delta$.*

Even tough it is crucial in the stochastic setting, it turns out that in the adversarial setting the optimistic bonus $b_t$ is not really needed. Indeed, as we will show in the following, we are interested in obtaining no-regret with respect to the set $\mathcal{X}_\varnothing^\star$ which is obtained via interpolation of points in $\mathcal{X}$ and the strictly feasible actions $a^\varnothing$. Let $x^\varnothing$ be such that $x^\varnothing(a^\varnothing) = 1$ and $x^\varnothing(a) = 0$ for all $a \neq a^\varnothing$. Formally:

$$\mathcal{X}_\varnothing^\star := \frac{1}{1+\rho}\{x^\varnothing\} + \frac{\rho}{1+\rho}\mathcal{X}, \tag{2}$$

where $A + B$ is the Minkowski sum between sets and $\alpha A$ indicates the set that contains each element of $A$ multiplied by $\alpha$.[4] The following theorem proves that $\mathcal{X}_\varnothing^\star \subseteq \widehat{\mathcal{X}}_t$ for all $t$.

**Theorem 5.2.** *In the adversarial setting, it holds $\mathcal{X}_\varnothing^\star \subseteq \widehat{\mathcal{X}}_t$ for all $t \in [\![T]\!]$.*

Notice that having no-regret with respect to the set $\mathcal{X}_\varnothing^\star$ is not sufficient to achieve no-regret in the adversarial setting. Nonetheless, we will show that this is sufficient to guarantee no-$\alpha$-regret, for $\alpha = \rho/1+\rho$ with respect to any strategy $x \in \Delta_K$.

## 5.2 How to set the weights

We focus on the design of estimators $\hat{g}_t^{(i)}$ that are good approximations of the real functions $g_t^{(i)}$. Algorithm 1 computes the estimators $\hat{g}_t^{(i)}$ by using a weighted mean of all past observations:

$$\hat{g}_t^{(i)}(a) = \sum_{\tau \in \mathcal{T}_{t-1,a}} w_{t,a}^{(i)}(\tau) g_\tau^{(i)}(a) \quad \forall t \in [\![T]\!], a \in [\![K]\!], i \in [\![M]\!].$$

However, to simplify the exposition, we use the following equivalence between online gradient descent (**OGD**) on quadratic losses $\hat{g}_t^{(i)}(a_t) \mapsto \frac{1}{2}\left(g_t^{(i)}(a_t) - \hat{g}_t^{(i)}(a_t)\right)^2$ and weighted means. In particular this equivalence is realized by observing that such loss has gradient $g_t^{(i)}(a_t) - \hat{g}_t^{(i)}(a_t)$.

**Lemma 5.3.** *Given any sequence $\{y_t\}_{t \in [\![T]\!]}$ such that $y_1 = 0$ and any sequence of learning rates $\{\eta_t\}_{t \in [\![T]\!]}$ such that $\eta_1 = 1$, let $\{\hat{y}_t\}_{t \in [\![T]\!]}$ be the estimator updated as:*

$$\hat{y}_{t+1} = \hat{y}_t + \eta_t(y_t - \hat{y}_t).$$

*Then, it holds that $\hat{y}_t = \sum_{\tau=1}^{t-1} y_\tau w_t(\tau)$ where $w_t(\tau) = \eta_\tau \prod_{k=\tau+1}^{t-1}(1 - \eta_k)$. Moreover, $\sum_{\tau=1}^{t-1} w_t(\tau) = 1$ for any $t \geq 2$.*

Clearly, in the **OGD** interpretation of our update, we only update $\hat{g}_t^{(i)}(a)$ only when $a_t = a$, and thus we only need to define learning rates for action $a$ for the times $t$ in which $a_t = a$. Based on this observation, we are going to update $\hat{g}_t^{(i)}(a)$ as

$$\begin{cases} \hat{g}_{t+1}^{(i)}(a_t) = \hat{g}_t^i(a_t) + \eta_t^{(i)}(a_t)\left(g_t^{(i)}(a_t) - \hat{g}_t^i(a_t)\right) \\ \hat{g}_{t+1}^{(i)}(a) = \hat{g}_t^{(i)}(a) & \forall a \neq a_t. \end{cases}$$

---

[4] Formally Minkowski sum between sets $A + B$ is defined as $A + B := \{a + b : a \in A, b \in B\}$.

Thus, given an action $a \in [\![K]\!]$ and a time $t \in [\![T]\!]$ the corresponding weights $\{w_{t,a}^{(i)}(\tau)\}_{\tau \in \mathcal{T}_{t-1,a}}$ are:

$$w_{t,a}^{(i)}(\tau) = \eta_\tau^{(i)}(a) \prod_{k \in \mathcal{T}_{t-1,a}:k>\tau} (1 - \eta_k^{(i)}(a)) \quad \forall \tau \in \mathcal{T}_{t-1,a}$$

We now proceed to give two notable examples on how to instantiate the learning rates and recover commonly used estimators such as the empirical mean and the exponentially weighted mean.[5]

**Proposition 5.4.** *If $\eta_t^{(i)}(a_t) = \frac{1}{n_t(a_t)}$ for each $\tau \in \mathcal{T}_{t-1,a}$, then $w_{t,a}^{(i)}(\tau) = \frac{1}{n_{t-1}(a)}$ and we recover the empirical mean estimator for $\hat{g}_t^{(i)}(a) = \frac{1}{n_{t-1}(a)} \sum_{\tau \in \mathcal{T}_{t-1,a}} g_\tau^{(i)}(a)$.*

**Proposition 5.5.** *If $\eta_t^{(i)}(a_t) = \eta$ then*

$$w_{t,a}^{(i)}(\tau) = \eta(1-\eta)^{|\{k \in \mathcal{T}_{t-1,a}:k>\tau\}|}$$

*for each $\tau \in \mathcal{T}_{t-1,a}$ and we recover an exponentially weighted average estimator for $\hat{g}_t^{(i)}(a)$.*

As it will turns out, these are the two extreme cases that we want to interpolate between. Indeed, the empirical mean estimator is particularly effective in the stochastic case but ineffective in the adversarial case, while the converse happens with the exponentially weighted estimator.

Now, we show that the OGD interpretation is particularly useful to bounds the violations suffered by the algorithm. First, we define the violations in an interval $[t_1, t_2] := \{t \in [\![T]\!] : t_1 \le t \le t_2\}$ as:

$$V_{[t_1,t_2]}^{(i)} = \sum_{t=t_1}^{t_2} g_t^{(i)}(a_t).$$

Then, in the following lemma we show that the violations in the interval are related to the variation of the estimates $\hat{g}_t^{(i)}(a)$.

**Theorem 5.6.** *Given an interval $[t_1, t_2] \subseteq [\![T]\!]$, an $i \in [\![m]\!]$, and a $\delta > 0$, with probability at least $1 - \delta$ it holds:*

$$V_{[t_1,t_2]}^{(i)} \le \sum_{a \in [\![K]\!]} \sum_{\tau \in \mathcal{T}_{t_2,a} \cap [t_1,t_2]} \frac{1}{\eta_\tau^{(i)}(a)} \left( \hat{g}_{\tau+1}^{(i)}(a) - \hat{g}_\tau^{(i)}(a) \right) + \sum_{\tau=t_1}^{t_2} \langle x_\tau, b_\tau \rangle + 4\sqrt{(t_2 - t_1) \log(1/\delta)}.$$

By a simple telescoping argument, we have the following corollary, which holds whenever the learning rates are non-increasing within a time interval. Let $\ell(a, [t_1, t_2])$ be the last rounds in the interval $[t_1, t_2]$ in which action $a$ is played.

**Corollary 5.7.** *Given an interval $[t_1, t_2] \subseteq [\![T]\!]$, a $i \in [\![m]\!]$, and a $\delta > 0$, assume that for any $a \in [\![K]\!]$ it holds $\eta_\tau^{(i)}(a) \ge \eta_{\tau'}^{(i)}(a) \ \forall \tau < \tau' \in \mathcal{T}_{t_2,a} \cap [t_1, t_2]$. Then, with probability at least $1 - \delta$ it holds:*

$$V_{[t_1,t_2]}^{(i)} \le \sum_{a \in [\![K]\!]} \frac{2}{\eta_{\ell(a,[t_1,t_2])}^{(i)}(a)} + \sum_{\tau=t_1}^{t_2} \langle x_\tau, b_\tau \rangle + 4\sqrt{(t_2 - t_1) \log(1/\delta)}.$$

Corollary 5.7 shows how to bound the violation as a function of the learning rates $\eta_t^{(i)}$ and the bonus terms $b_\tau$. The following lemma shows how to bound the second term of the violations depending on the structure of the bonus terms.

**Lemma 5.8.** *Given a $c > 0$, an $\alpha \in (0, 1)$, a $t \in [\![T]\!]$, and a $\delta > 0$, let $b_t(a) = \frac{c}{n_t(a)^\alpha}$ for all $a \in [\![K]\!]$. Then, with probability at least $1 - \delta$, it holds:*

$$\sum_{\tau=1}^{t} \langle x_\tau, b_\tau \rangle \le \frac{c}{1-\alpha} K^\alpha t^{1-\alpha} + 4\sqrt{t \log(1/\delta)}.$$

In this section, we saw how the choice of the learning rates of the estimator affects the estimators. In the following section, we will see how to *adaptively* set those learning rates to handle both stochastic and adversarial settings.

---

[5]The proof of the first proposition can be found in Appendix D, while the proof of the second is straightforward and thus it is omitted.

# 6 Adaptive learning rates

The previous section highlights the main difficulties of obtaining best-of-both-world algorithms: we need to set the weights $w_{t,a}^{(i)}$ (or equivalently - by Lemma 5.3 - the learning rates $\eta_t^{(i)}(a_t)$) and the optimistic bonuses $b_t$ so that they meet, at the same time, the requirements needed by the stochastic and the adversarial settings.

We start presenting two possible choices and show that they fail either in the stochastic or the adversarial setting. Then, we show how adaptive learning rates combine the strengths of both approaches. The first, natural, choice of setting the learning rate is to use an exponentially weighted estimator, i.e., choose $\eta_t^{(i)}(a_t) = 1/\sqrt{T}$. With this choice, we can apply a weighted version of Azuma-Hoeffding inequality and find that $|\hat{g}_t^{(i)}(a) - \bar{g}^{(i)}(a)| \in \widetilde{O}\left(n_t(a)^{-1/4}\right)$, with high probability. Thus, as discussed in Section 5.1, we would need to define $b_t(a) \in \widetilde{O}\left(n_t(a)^{-1/4}\right)$, which, by Corollary 5.7 and Lemma 5.8 would imply a suboptimal $\tilde{O}(T^{3/4})$ rate for the violations.

The second option is to set $\eta_t^{(i)}(a_t) = 1/n_t(a_t)$. In the stochastic setting, we have an optimal rate of concentration of the terms $|\hat{g}_t^{(i)}(a) - \bar{g}^{(i)}(a)| \in \widetilde{O}\left(n_t(a)^{-1/2}\right)$ as, by Proposition 5.4, this is equivalent to compute the empirical mean. However, this second option fails disastrously in the adversarial setting as highlighted in Corollary 5.7, where the first component of the violations becomes linear in $T$. Intuitively, a learning rate of order $1/n_t(a)$ makes the update of the estimates too slow when the underlying constraints change, as it does happen in the adversarial setting.

This trade-off forces us to employ *adaptive learning rates*. Our idea is to use learning rates of the order $1/n_t(a)$ with an adaptive multiplicative term that depends on the current violation of the constraint. Formally, we use learning rates:

$$\eta_t^{(i)}(a_t) := \frac{1}{n_t(a)}\left(1 + \Gamma_t^{(i)}\right),$$

where $\Gamma_t^{(i)}$ is a bonus term defined as

$$\Gamma_t^{(i)} := \left[V_{t-1}^{(i)} - 21\sqrt{Kt\log(1/\delta_2)}\right]_0^{21\sqrt{Kt\log(1/\delta_2)}},$$

and $[x]_a^b := \min(\max(x, a), b)$ is the clipping of $x$ between $a$ and $b$. Moreover, we set the exploration bonus as

$$b_t(a) = \sqrt{\frac{2\log(2/\delta_2)}{n_{t-1}(a)}}.$$

The following theorem shows that such approach guarantees $\widetilde{O}(\sqrt{KT})$ violations in both adversarial and stochastic settings.

**Theorem 6.1.** *Both in the stochastic and the adversarial setting, with probability at least* $1 - 2mT^2\delta_2$ *it holds that*

$$V_t \leq 53\sqrt{Kt\log(2/\delta_2)} \quad \forall t \in [\![T]\!].$$

The previous theorem shows that this choice of learning rates is sufficient to guarantee optimal bounds on the violations. However, to achieve this result we are setting $b_t(a) \in \widetilde{O}(n_t(a)^{-1/2})$. As we showed in theorem 5.1, this requires a concentration on the estimates $|\hat{g}_t^{(i)}(a) - \bar{g}_t^{(i)}(a)|$ of the same magnitude (in the stochastic setting). This is crucially needed to ensure that the regret minimizer $\mathcal{R}$ provides the desired guarantees and that the event $\mathcal{C}$ defined in Section 3 actually holds with high probability.

**Lemma 6.2.** *In the stochastic setting, with probability at least* $1 - 5mKT\delta_2$, *it holds that:*

$$|\hat{g}_t^{(i)}(a) - \bar{g}_t^{(i)}(a)| \leq b_t(a) \quad \forall a \in [\![K]\!], t \in [\![T]\!], i \in [\![m]\!]$$

The proof of the previous result relies on the fact that in the stochastic case the bonus $\Gamma_t^{(i)}$ does not "kick in" ensuring that $\eta_t^{(i)}(a) = 1/n_t(a)$. Thus, $\hat{g}_t^{(i)}$ is the empirical average of past observations. The previous result, together with Theorem 5.1 proves the following corollary.

**Corollary 6.3.** *In the stochastic setting, with probability at least $1 - 5mKT\delta_2$, it holds that $\mathcal{X}^\star \in \widehat{\mathcal{X}}_t$ for all $t \in [\![T]\!]$.*

This proves that the clean event $\mathcal{C}$ holds with high probability, as promised in Section 3.

## 7 Putting everything together

Now, we have everything in place to easily prove the our main theorems. First, we define the parameters $\delta_1 = \delta_1(\epsilon)$ and $\delta_2 = \delta_2(\epsilon)$ in order to guarantee that our theorems hold with probability at least $1 - \epsilon$. In particular, we set $\delta_1(\epsilon) = \epsilon/2$, where we recall that $\delta_1$ is the parameter used to set $\beta$ and $\gamma$ in Algorithm 2, and $\delta_2(\epsilon) = \epsilon/(14mKT^2)$, where $\delta_2$ is used to set the optimistic bonus and learning rate of Algorithm 1.

In the stochastic setting, the violation guarantees directly follow from Theorem 6.1, while the regret guarantee follows by combining Theorem 4.1 and Corollary 6.3. Formally:

**Theorem 7.1.** *In the stochastic setting, for any $\epsilon > 0$ Algorithm 1 guarantees that with probability at least $1 - \epsilon$:*

$$R_T \leq 4\sqrt{KT\log(2K/\epsilon)} \quad and \quad V_t \leq 53\sqrt{Kt\log(28mKT^2/\epsilon)} \quad \forall t \in [\![T]\!].$$

Now, we turn to the adversarial setting. Theorem 6.1 guarantee $\widetilde{O}(\sqrt{T})$ violations even with adversarial constraint, while the regret guarantees follows by combining Theorem 5.2 and Theorem 4.1

**Theorem 7.2.** *In the adversarial setting, for any $\epsilon > 0$ Algorithm 1 guarantees that with probability at least $1 - \epsilon$:*

$$\alpha\text{-}R_T \leq 4\sqrt{KT\log(2K/\epsilon)} \quad and \quad V_t \leq 53\sqrt{Kt\log(28mKT^2/\epsilon)} \quad \forall t \in [\![T]\!],$$

*where $\alpha = \rho/(1+\rho)$.*

Note that in both settings, the regret upper bound is of order $\widetilde{O}(\sqrt{KT})$ and it is independent from the number of constraints $m$, while the violations are of order $\widetilde{O}(\sqrt{KT\log(m)})$ and depend only logarithmically on $m$. This is in contrast to the other best-of-both-world algorithms for bandits with long term constraints, based on primal-dual methods, in which both the regret and the violations depends polynomially in $m$.

Another interesting characteristic of our methodology is that we guarantee an anytime bound on the constraint violation. Indeed, this matches the guarantees provided by the most recent primal-dual methods [11, 1] that, however, require weakly-adaptive underlying regret minimizers.

### 7.1 Convergence rate in the stochastic setting

To conclude, we point to a nice byproduct of our analysis. In the stochastic setting, we can easily prove a sort of "convergence rate" of $x_t$ to the set $\mathcal{X}^\star$. Formally, we can prove that *positive violations* are bounded by $\widetilde{O}(\sqrt{Kt\log m})$ as long as we consider expected violations. Let us define $x^+ := \max(x, 0)$ and

$$\mathcal{V}_t^+ := \max_{i \in [\![m]\!]} \sum_{\tau=1}^{t} \left[\langle x_\tau, \bar{g}_\tau^{(i)}\rangle\right]^+.$$

Then, we can state the following theorem:

**Theorem 7.3.** *Algorithm 1, in the stochastic setting, guarantees that with probability at least $1 - \epsilon$, it holds that:*

$$\mathcal{V}_t^+ \leq 16\sqrt{Kt\log(28mKT^2/\epsilon)} \quad \forall t \in [\![T]\!].$$

Intuitively, our result shows that our algorithm plays only a sublinear number of times "far" from the set $\mathcal{X}^\star$, or that our algorithm plays a linear number of times "close" to the set $\mathcal{X}^\star$. This is a much stronger result then just guaranteeing that $V_T$ is sublinear, as in that case it might be a linear number of times the algorithm plays "far" from $\mathcal{X}^\star$ as long as it plays strictly inside of $\mathcal{X}^\star$ often enough.

## Acknowledgments

MB, MC, AC, FF are partially supported by the FAIR (Future Artificial Intelligence Research) project PE0000013, funded by the NextGenerationEU program within the PNRR-PE-AI scheme (M4C2, investment 1.3, line on Artificial Intelligence). FF is also partially supported by ERC Advanced Grant 788893 AMDROMA "Algorithmic and Mechanism Design Research in Online Markets", and PNRR MUR project IR0000013-SoBigData.it. MC is also partially supported by the EU Horizon project ELIAS (European Lighthouse of AI for Sustainability, No. 101120237). AC is partially supported by MUR - PRIN 2022 project 2022R45NBB funded by the NextGenerationEU program.

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

# A Further Related Works

**Best-of-Both-Worlds.** A long line of work has investigated Best-of-Both-Worlds algorithms for bandits without constraints. These algorithms aim to achieve an instance-dependent logarithmic regret bound in stochastic environments, while also ensuring the worst-case $\Theta(\sqrt{T})$ regret bound that characterizes the adversarial settings [14, 4, 27, 26, 29, 30]. Although our focus is on the generation model of the *constraints*, our motivation in this paper is affine: retaining the best of the stochastic (sublinear regret with respect to the optimal dynamic policy) and adversarial world (tight competitive ratio with respect to the adversarial benchamrk). Furthermore, our idea of setting an adaptive learning rate that forces the learning algorithm to interpolate between an adversarial and a stochastic routine is reminiscent of some of the techniques adopted in, e.g., Bubeck and Slivkins [14].

**Bandits with Knapsacks.** The (stochastic) BwK problem, where the rewards $f_t$ as well as the $g_t^i$ are drawn i.i.d. from a non-negative distribution (so that the budget available for each resource can only decrease over time) is formally introduced and solved in Badanidiyuru et al. [6] (see also its journal version [7]). Agrawal and Devanur [2] studies a more general stochastic setting, which subsumes knapsack and exhibit optimal guarantees via *optimism in the face of uncertainty* [see also 3]. Moving to the adversarial BwK problem (which corresponds to our model when the $g_t^i$ are all non-negative), an optimal solution is proposed in Immorlica et al. [20] [see also 21]; there, the authors propose the LagrangeBwK framework, which has a natural interpretation: arms can be thought of as primal variables, and resources as dual variables. The framework works by setting up a repeated two-player zero-sum game between a primal and a dual player, and by showing convergence to a Nash equilibrium of the expected Lagrangian game. Differently from the stochastic version, the adversarial BwK does not admit no-regret algorithms, but $\Theta(\log T)$ competitive ratio. In a subsequent work, [22] provides a new analysis obtaining a $O(\log m \log T)$ competitive ratio, which is optimal both in the time horizon $T$ and in the number of resources $m$ (and improves on the $O(m \log T)$ of Immorlica et al. [20, 21]). In the special case in which budgets are $\Omega(T)$, Castiglioni et al. [15] further improves the competitive ratio to $1/\rho$ where $\rho$ is the per-iteration budget.

**More general constraints.** Castiglioni et al. [15] studies a setting with general constraints, and show how to adapt the LagrangeBwK framework to obtain best-of-both-worlds guarantees when Slater's parameter is known a priori. Similar guarantees are also provided, in the stochastic setting, by Slivkins et al. [28], which then extend the results to the contextual model. Finally, Castiglioni et al. [17] introduces the use of weakly adaptive regret minimizers within the LagrangeBwK framework, and provides guarantees in the specific case of one budget constraint and one return-on-investments constraint.

**Other related works.** In an effort to bridge the results for adversarial and stochastic BwK, Fikioris and Tardos [18] investigates a data generation model that interpolate between the fully stochastic and the fully adversarial setting, depending on the magnitude of fluctuations in expected rewards and resources consumption across rounds. A similar effort is undertaken in Liu et al. [24], that study a non-stationary setting and provide no-regret guarantees against the best dynamic policy through a UCB-based algorithm. A recent line of work also investigates the natural situation where resources can be replenished in certain rounds (as also captured in our model) [23, 13, 12]. Finally, a related line of works is the one on online allocation problems with fixed per-iteration budget, where the input pair of reward and costs is observed *before* the learner makes a decision [10, 8].

# B Proofs omitted from Section 4

**Theorem 4.1.** *Let $x_t$ be selected accordingly to Algorithm 2 run with arbitrary sequence of convex sets $\widehat{\mathcal{X}}_t \subseteq \Delta_K$ with $\gamma = \frac{\beta}{2}$ and $\beta = \sqrt{\frac{\log(K/\delta_1)}{KT}}$. Then, with probability at least $1 - \delta_1$ it holds that*

$$\sum_{t \in [\![T]\!]} \langle f_t, x \rangle - f_t(a_t) \le 4\sqrt{KT \log(K/\delta_1)}, \qquad \forall x \in \bigcap_{t \in [\![T]\!]} \widehat{\mathcal{X}}_t.$$

*Proof.* Let us define the negative entropy for a vector $x \in \mathbb{R}_{\geq 0}^K$ as:

$$\Psi(x) := \sum_{a \in [\![K]\!]} x(a) \left( \log(x(a)) - 1 \right)$$

and the Bregman divergence using $\Psi$ can be written as

$$B(x||y) := \Psi(x) - \Psi(y) - \langle \nabla \Psi(y), x - y \rangle.$$

For the Bregman divergence it holds the following:

**Claim B.1** ([19])**.** For any $z_1, z_2,$ and $z_3$, it holds:

$$B(z_1||z_2) + B(z_2||z_3) - B(z_1||z_3) = \langle z_1 - z_2, \nabla \Psi(z_3) - \nabla \Psi(z_2) \rangle.$$

Moreover, given $z$, define $z' = \arg\min_{\bar{z} \in \mathcal{K}} B(\bar{z}||z)$. Then:

$$B(\tilde{z}||z') \leq B(z'||z) + B(\tilde{z}||z') \leq B(\tilde{z}||z) \quad \forall \tilde{z} \in \mathcal{K}.$$

At this point, is more convenient to work with losses rather then rewards. Define $\ell_t(a) := 1 - f_t(a)$ and $\hat{\ell}_t(a) := 1 - \hat{f}_t(a)$. Note that:

$$\hat{\ell}_t(a) = 1 - \hat{f}_t(a) = \begin{cases} 0 & \text{if } a \neq a_t \\ \frac{1 - f_t(a)}{x_t(a) + \gamma} & \text{if } a = a_t. \end{cases}$$

Then, it is easy to verify that $\nabla \Psi(x) = \log(x)$ in which $\log(x)$ has to be interpreted to be applied entry-wise. Simple calculations also show that $\beta \hat{\ell}_t = \log(x_t) - \log(\hat{x}_{t+1})$. Thus, we can apply Claim B.1 with $z_1 = x, z_2 = x_t$ and $z_3 = \hat{x}_{t+1}$ and this gives us the following:

$$\beta \langle x_t - x, \hat{\ell}_t \rangle = B(x||x_t) + B(x_t||\hat{x}_{t+1}) - B(x||\hat{x}_{t+1}). \tag{3}$$

Moreover using the second part of Claim B.1 in which $z = \hat{x}, z' = x_t, \tilde{z} = x,$ and $\mathcal{K} = \widehat{\mathcal{X}}_t$, we can conclude that $B(x||x_t) \leq B(x||\hat{x}_t)$. Notice that here we use $x \in \widehat{\mathcal{X}}_t$ for each $t$. Then, we have the following chain of inequalities:

$$\beta \sum_{t \in [\![T]\!]} \langle x_t - x, \hat{\ell}_t \rangle = \sum_{t \in [\![T]\!]} B(x||x_t) + B(x_t||\hat{x}_{t+1}) - B(x||\hat{x}_{t+1}) \qquad \text{(By Equation (3))}$$

$$= B(x||x_1) - B(x||\hat{x}_{T+1}) + \sum_{t=2}^{T-1} \left( B(x||x_t) - B(x||\hat{x}_t) \right) + \sum_{t \in [\![T]\!]} B(x_t||\hat{x}_{t+1})$$

$$\leq B(x||x_1) + \sum_{t \in [\![T]\!]} B(x_t||\hat{x}_{t+1}) \quad \text{($B$ is non-negative and $B(\cdot||x_t) \leq B(\cdot||\hat{x}_t)$)}$$

$$= B(x||x_1) + \sum_{t \in [\![T-1]\!]} B(x_t||\hat{x}_{t+1})$$

Combining the two we can find that:

$$\beta \sum_{t \in [\![T]\!]} \langle x_t - x, \hat{\ell}_t \rangle \leq \sum_{t \in [\![T]\!]} \left[ B(x||\hat{x}_t) + B(x_t||\hat{x}_{t+1}) - B(x||\hat{x}_{t+1}) \right] \tag{4}$$

$$\leq B(x||x_1) + \sum_{t \in [\![T]\!]} B(x_t||\hat{x}_{t+1}) \tag{5}$$

Now we analyze the term $B(x_t||\hat{x}_{t+1})$.

$$
\begin{aligned}
B(x_t||\hat{x}_{t+1}) &\leq B(x_t||\hat{x}_{t+1}) + B(\hat{x}_{t+1}||x_t) \\
&= \langle x_t - \hat{x}_{t+1}, \nabla\Psi(x_t) - \nabla\Psi(\hat{x}_{t+1}) \rangle && \text{(Definition of } B(\cdot||\cdot)) \\
&= \beta\langle x_t - \hat{x}_{t+1}, \hat{\ell}_t \rangle && (\nabla\Psi(x) = \log(x) \text{ and } \beta\hat{\ell}_t = \log(x_t) - \log(\hat{x}_{t+1})) \\
&= \beta \sum_{a\in[\![K]\!]} x_t(a)(1 - e^{-\beta\hat{\ell}_t(a)})\hat{\ell}_t(a) \\
&\leq \beta^2 \sum_{a\in[\![K]\!]} x_t(a)\hat{\ell}_t(a)^2 && (1 - e^{-x} \leq x) \\
&\leq \beta^2 \sum_{a\in[\![K]\!]} \frac{1 - f_t(a)}{x_t(a) + \gamma} x_t(a)\hat{\ell}_t(a) \\
&\leq \beta^2 \sum_{a\in[\![K]\!]} \hat{\ell}_t(a),
\end{aligned}
$$

where, in the last inequality, we use that $x_t(a)/(x_t(a)+\gamma)$ is at most 1. Thus, by choosing $x_1(a) = 1/K$ for all $a$, we have that $B(x||x_1) \leq \log(K)$ and thus:

$$
\sum_{t\in[\![T]\!]} \langle x_t - x, \hat{\ell}_t \rangle \leq \frac{\log(K)}{\beta} + \beta \sum_{t\in[\![T]\!]}\sum_{a\in[\![K]\!]} \hat{\ell}_t(a) \tag{6}
$$

Form [25, Corollary 1] we know that with probability at least $1 - \delta_1$ we have:

$$
\sum_{t\in[\![T]\!]} \hat{\ell}_t(a) - (1 - f_t(a)) \leq \frac{\log(K/\delta_1)}{2\gamma} \quad \forall a \in [\![K]\!]. \tag{7}
$$

Moreover, it is easy to verify that:

$$
\begin{aligned}
1 - f_t(a_t) &= \sum_{a\in[\![K]\!]} \mathbb{I}(a_t = a)(1 - f_t(a))\frac{x_t(a) + \gamma}{x_t(a) + \gamma} \\
&= \sum_{a\in[\![K]\!]} \hat{\ell}_t(a)x_t(a) + \gamma \sum_{a\in[\![K]\!]} \frac{\hat{\ell}_t(a)\mathbb{I}(a_t = a)}{x_t(a) + \gamma} \\
&= \langle x_t, \hat{\ell}_t \rangle + \gamma \sum_{a\in[\![K]\!]} \hat{\ell}_t(a) \tag{8}
\end{aligned}
$$

The regret is with probability at least $1 - \delta_1$:

$$
\begin{aligned}
\sum_{t\in[\![T]\!]} &[\langle x, f_t \rangle - f_t(a_t)] \\
&= \sum_{t\in[\![T]\!]} [(1 - f_t(a_t)) - (1 - \langle x, f_t \rangle)] \\
&= \sum_{t\in[\![T]\!]} [(1 - f_t(a_t)) - \langle x, \hat{\ell}_t \rangle] + \sum_{t\in[\![T]\!]} [\langle x, \hat{\ell}_t \rangle - (1 - \langle x, f_t \rangle)] \\
&\leq \sum_{t\in[\![T]\!]} \langle x_t - x, \hat{\ell}_t \rangle + \sum_{t\in[\![T]\!]} [\langle x, \hat{\ell}_t \rangle - (1 - \langle x, f_t \rangle)] + \gamma \sum_{t\in[\![T]\!]}\sum_{a\in[\![K]\!]} \hat{\ell}_t(a) && \text{(Equation (8))} \\
&\leq \frac{\log(K)}{\beta} + \frac{\log(K/\delta_1)}{2\gamma} + (\gamma + \beta)\sum_{t\in[\![T]\!]}\sum_{a\in[\![K]\!]} \hat{\ell}_t(a) && \text{(Equation (6) and Equation (7))} \\
&\leq \frac{\log(K)}{\beta} + \frac{\log(K/\delta_1)}{2\gamma} + (\gamma + \beta)\left[\sum_{t\in[\![T]\!]}\sum_{a\in[\![K]\!]} (1 - f_t(a)) + K\frac{\log(K/\delta_1)}{2\gamma}\right] \\
&\leq \frac{\log(K)}{\beta} + \frac{\log(K/\delta_1)}{2\gamma} + (\gamma + \beta)KT + (\gamma + \beta)K\frac{\log(K/\delta_1)}{2\gamma} \\
&= \frac{\log(K)}{\beta} + \frac{\log(K/\delta_1)}{\beta} + 2\beta KT + 2K\log(K/\delta_1)
\end{aligned}
$$

where in the last inequality we used that $\beta = 2\gamma$. By taking $\beta = \sqrt{\frac{\log(K/\delta_1)}{KT}}$ we obtain, that with probability at least $1 - \delta_1$:

$$\sum_{t \in [\![T]\!]} [\langle x, f_t \rangle - f_t(a_t)] \leq 4\sqrt{KT \log(K/\delta_1)},$$

as desired. $\qquad\square$

## C Proofs omitted from Section 5.1: How to set the optimistic bonus

**Theorem 5.1.** *Consider the stochastic setting. Given any $\delta > 0$, let $b_t(a)$ be such that with probability at least $1 - \delta$ it holds:*

$$|\hat{g}_t^{(i)}(a) - \bar{g}^{(i)}(a)| \leq b_t(a) \quad \forall t \in [\![T]\!], i \in [\![m]\!], a \in [\![K]\!].$$

*Then, it holds $\mathcal{X}^\star \subseteq \cap_{t \in [\![T]\!]} \widehat{\mathcal{X}}_t$ with probability at least $1 - \delta$.*

*Proof.* In the following, we assume that the condition in the statement of the theorem holds. Hence, our result with hold with probability $1 - \delta$ as promised. Let $x \in \mathcal{X}_i^\star$. Consider a $t \in [\![T]\!]$ and an $i \in [\![m]\!]$. Then, consider the following inequalities:

$$
\begin{aligned}
\langle x, \hat{g}_t^{(i)} \rangle &= \langle x, \hat{g}_t^{(i)} - \bar{g}^{(i)} \rangle + \langle x, \bar{g}^{(i)} \rangle \\
&\leq \langle x, \hat{g}_t^{(i)} - \bar{g}^{(i)} \rangle && (x \in \mathcal{X}_i^\star) \\
&= \sum_{a \in [\![K]\!]} x(a)(\hat{g}_t^{(i)}(a) - \bar{g}^{(i)}(a)) \\
&\leq \langle x, b_t \rangle.
\end{aligned}
$$

Thus, $\langle x, \hat{g}_t^{(i)} - b_t \rangle \leq 0$ which, by definition, proves that $x \in \widehat{\mathcal{X}}_t^{(i)}$. This concludes the proof. $\qquad\square$

**Theorem 5.2.** *In the adversarial setting, it holds $\mathcal{X}_\varnothing^\star \subseteq \widehat{\mathcal{X}}_t$ for all $t \in [\![T]\!]$.*

*Proof.* In the adversarial setting, by Equation (1) we have that

$$g_t^{(i)}(a^\varnothing) \leq -\rho,$$

for all $t \in [\![T]\!]$ and constraint $i \in [\![m]\!]$. Moreover, for each $t \in [\![T]\!]$, $i \in [\![m]\!]$, and $a \in [\![K]\!]$, it holds

$$\hat{g}_t^{(i)}(a) = \sum_{\tau \in \mathcal{T}_{t-1,a}} w_{t,a}^{(i)}(\tau) \, g_\tau^{(i)}(a)$$

and $\sum_{\tau \in \mathcal{T}_{t-1,a}} w_{t,a}^{(i)}(\tau) = 1$. Then, for all $t \in [\![T]\!]$ and constraint $i \in [\![m]\!]$, $\hat{g}_t^{(i)}(a^\varnothing) \leq -\rho$ and $\hat{g}_t^{(i)}(a) \leq 1$ for each $a \neq a^\varnothing$. [6] Thus, we can consider the following inequalities for any $\tilde{x} \in \mathcal{X}_\varnothing^\star$:

$$
\begin{aligned}
\langle \tilde{x}, \hat{g}_t^{(i)} \rangle &= \frac{1}{1+\rho}\hat{g}_t^{(i)}(a^\varnothing) + \frac{\rho}{1+\rho}\langle x, \hat{g}_t^{(i)} \rangle \\
&\leq \frac{1}{1+\rho}(-\rho) + \frac{\rho}{1+\rho} \\
&\leq 0,
\end{aligned}
$$

thus proving that $\tilde{x} \in \widehat{\mathcal{X}}_t$. $\qquad\square$

---

[6] Notice that these inequalities hold only for action played at least one time. Otherwise, similar inequalities continue to be true thanks to the optimistic bonus $b_t$.

# D  Proofs omitted from Section 5.2: How to set the weights

**Lemma 5.3.** *Given any sequence $\{y_t\}_{t\in[\![T]\!]}$ such that $y_1 = 0$ and any sequence of learning rates $\{\eta_t\}_{t\in[\![T]\!]}$ such that $\eta_1 = 1$, let $\{\hat{y}_t\}_{t\in[\![T]\!]}$ be the estimator updated as:*

$$\hat{y}_{t+1} = \hat{y}_t + \eta_t(y_t - \hat{y}_t).$$

*Then, it holds that $\hat{y}_t = \sum_{\tau=1}^{t-1} y_\tau w_t(\tau)$ where $w_t(\tau) = \eta_\tau \prod_{k=\tau+1}^{t-1}(1 - \eta_k)$. Moreover, $\sum_{\tau=1}^{t-1} w_t(\tau) = 1$ for any $t \geq 2$.*

*Proof.* The first part of the statement is trivial as it can be easily checked that:

$$\hat{y}_t = \sum_{\tau=1}^{t-1} y_\tau \left( \eta_\tau \prod_{k=\tau+1}^{t-1}(1 - \eta_k) \right).$$

Then, we prove the second part of the lemma by induction on $t$. The base case holds trivially as $w_2(1) = \eta_1 = 1$. Moreover, assuming $\sum_{\tau=1}^{t-2} w_\tau^t = 1$, it holds:

$$\sum_{\tau=1}^{t-1} w_t(\tau) = \sum_{\tau=1}^{t-2} w_{t-1}(\tau)(1 - \eta_{t-1}) + w_t(t-1) = (1 - \eta_{t-1}) + \eta_{t-1} = 1,$$

where in the second-to-last equality we use the inductive hypothesis. This concludes the proof. $\qquad\square$

**Proposition 5.4.** *If $\eta_t^{(i)}(a_t) = \frac{1}{n_t(a_t)}$ for each $\tau \in \mathcal{T}_{t-1,a}$, then $w_{t,a}^{(i)}(\tau) = \frac{1}{n_{t-1}(a)}$ and we recover the empirical mean estimator for $\hat{g}_t^{(i)}(a) = \frac{1}{n_{t-1}(a)} \sum_{\tau \in \mathcal{T}_{t-1,a}} g_\tau^{(i)}(a)$.*

*Proof.* Consider an $a \in [\![K]\!]$, an $i \in [\![m]\!]$, and a $t \in [\![T]\!]$. Then, by applying Lemma 5.3 to the set of rounds $\mathcal{T}_{t-1,a}$ we have that:

$$w_{t,a}^{(i)}(\tau) = \frac{1}{n_\tau(a)} \prod_{k\in\mathcal{T}_{t-1,a}:k>\tau} \left(1 - \frac{1}{n_k(a)}\right) \quad \forall \tau \in \mathcal{T}_{t-1,a}.$$

Now, we show that

$$\prod_{k\in\mathcal{T}_{t-1,a}:k>\tau} \left(1 - \frac{1}{n_k(a)}\right) = \prod_{k\in\mathcal{T}_{t-1,a}:k>\tau} \frac{n_k(a) - 1}{n_k(a)}$$

$$= \prod_{j=n_\tau(a)+1}^{n_{t-1}(a)} \frac{j-1}{j}$$

$$= \frac{n_\tau(a)}{n_{t-1}(a)},$$

and thus $w_{t,a}^{(i)}(\tau) = \frac{1}{n_{t-1}(a)}$, as desired. $\qquad\square$

**Theorem 5.6.** *Given an interval $[t_1, t_2] \subseteq [\![T]\!]$, an $i \in [\![m]\!]$, and a $\delta > 0$, with probability at least $1 - \delta$ it holds:*

$$V_{[t_1,t_2]}^{(i)} \leq \sum_{a\in[\![K]\!]} \sum_{\tau\in\mathcal{T}_{t_2,a}\cap[t_1,t_2]} \frac{1}{\eta_\tau^{(i)}(a)} \left(\hat{g}_{\tau+1}^{(i)}(a) - \hat{g}_\tau^{(i)}(a)\right) + \sum_{\tau=t_1}^{t_2} \langle x_\tau, b_\tau \rangle + 4\sqrt{(t_2 - t_1)\log(1/\delta)}.$$

*Proof.* First, applying Lemma G.1, we have that with probability $1 - \delta$ it holds:

$$\sum_{\tau=t_1}^{t_2} \langle g_\tau^{(i)}, x_\tau \rangle \geq \sum_{\tau=t_1}^{t_2} g_\tau^{(i)}(a_\tau) - 4\sqrt{(t_2 - t_1)\log(1/\delta)} \qquad\qquad (9)$$

Consider the following chain of inequalities:

$$V_{[t_1,t_2]}^{(i)} = \sum_{\tau=t_1}^{t_2} g_\tau^{(i)}(a_\tau)$$

$$\leq \sum_{\tau=t_1}^{t_2} g_\tau^{(i)}(a_\tau) - \sum_{\tau=t_1}^{t_2} \langle \hat{g}_\tau^{(i)}, x_\tau \rangle + \sum_{\tau=t_1}^{t_2} \langle x_\tau, b_\tau \rangle \qquad (x_\tau \in \widehat{\mathcal{X}}_\tau)$$

$$\leq \sum_{\tau=t_1}^{t_2} \left( g_\tau^{(i)}(a_\tau) - \hat{g}_\tau^{(i)}(a_\tau) \right) + \sum_{\tau=t_1}^{t_2} \langle x_\tau, b_\tau \rangle + 4\sqrt{(t_2 - t_1)\log(1/\delta)} \qquad \text{(Equation (9))}$$

$$= \sum_{a \in \llbracket K \rrbracket} \sum_{\tau \in \mathcal{T}_{t_2,a} \cap [t_1,t_2]} (g_\tau^{(i)}(a) - \hat{g}_\tau^{(i)}(a)) + \sum_{\tau=t_1}^{t_2} \langle x_\tau, b_\tau \rangle + 4\sqrt{(t_2 - t_1)\log(1/\delta)}$$

$$= \sum_{a \in \llbracket K \rrbracket} \sum_{\tau \in \mathcal{T}_{t_2,a} \cap [t_1,t_2]} \frac{\hat{g}_{\tau+1}^{(i)}(a) - \hat{g}_\tau^{(i)}(a)}{\eta_\tau^{(i)}(a)} + \sum_{\tau=t_1}^{t_2} \langle x_\tau, b_\tau \rangle + 4\sqrt{(t_2 - t_1)\log(1/\delta)},$$

where the last equality follows by the definition of the update:

$$\hat{g}_{\tau+1}^{(i)}(a) = \left( 1 - \eta_\tau^{(i)}(a) \right) \hat{g}_\tau^{(i)}(a) + \eta_\tau^{(i)}(a) g_\tau^{(i)}(a) \quad \text{for } a = a_\tau.$$

This concludes the proof. $\qquad \square$

**Corollary 5.7.** *Given an interval $[t_1, t_2] \subseteq \llbracket T \rrbracket$, a $i \in \llbracket m \rrbracket$, and a $\delta > 0$, assume that for any $a \in \llbracket K \rrbracket$ it holds $\eta_\tau^{(i)}(a) \geq \eta_{\tau'}^{(i)}(a) \; \forall \tau < \tau' \in \mathcal{T}_{t_2,a} \cap [t_1, t_2]$. Then, with probability at least $1 - \delta$ it holds:*

$$V_{[t_1,t_2]}^{(i)} \leq \sum_{a \in \llbracket K \rrbracket} \frac{2}{\eta_{\ell(a,[t_1,t_2])}^{(i)}(a)} + \sum_{\tau=t_1}^{t_2} \langle x_\tau, b_\tau \rangle + 4\sqrt{(t_2 - t_1)\log(1/\delta)}.$$

*Proof.* We assume that Theorem 5.6 holds, and hence our statement holds with probability $1 - \delta$. Then, to prove the statement it is sufficient to show that

$$\sum_{a \in \llbracket K \rrbracket} \sum_{\tau \in \mathcal{T}_{t_2,a} \cap [t_1,t_2]} \frac{1}{\eta_\tau^{(i)}(a)} \left( \hat{g}_{\tau+1}^{(i)}(a) - \hat{g}_\tau^{(i)}(a) \right) \leq \sum_{a \in \llbracket K \rrbracket} \sum_{\tau \in \mathcal{T}_{t_2,a} \cap [t_1,t_2]} \frac{1}{\eta_{t_2}^{(i)}(a)}.$$

Fix any $a \in \llbracket K \rrbracket$, and let $k = |\mathcal{T}_{t_2,a} \cap [t_1, t_2]|$ be the number of times action $a$ is played in the interval $[t_1, t_2]$. Moreover, let $\tau(j)$ be the rounds in which action $a$ is played the $j$-th time in the interval $[t_1, t_2]$. Then:

$$\sum_{\tau \in \mathcal{T}_{t_2,a} \cap [t_1,t_2]} \frac{1}{\eta_\tau^{(i)}(a)} \left( \hat{g}_{\tau+1}^{(i)}(a) - \hat{g}_\tau^{(i)}(a) \right)$$

$$= \sum_{j \in \llbracket k-1 \rrbracket} \frac{1}{\eta_{\tau(j)}^{(i)}(a)} \left( \hat{g}_{\tau(j+1)}^{(i)}(a) - \hat{g}_{\tau(j)}^{(i)}(a) \right) + \frac{1}{\eta_{\tau(k)}^{(i)}(a)} \left( \hat{g}_{\tau(k)+1}^{(i)}(a) - \hat{g}_{\tau(k)}^{(i)}(a) \right)$$

$$\leq \sum_{j \in \llbracket k-1 \rrbracket} \left( \frac{1}{\eta_{\tau(j+1)}^{(i)}(a)} \hat{g}_{\tau(j+1)}^{(i)}(a) - \frac{1}{\eta_{\tau(j)}^{(i)}(a)} \hat{g}_{\tau(j)}^{(i)}(a) \right) + \frac{1}{\eta_{\tau(k)}^{(i)}(a)} \left( \hat{g}_{\tau(k)+1}^{(i)}(a) - \hat{g}_{\tau(k)}^{(i)}(a) \right)$$

$$= \frac{1}{\eta_{\tau(k)}^{(i)}(a)} \hat{g}_{\tau(k)+1}^{(i)}(a) - \frac{1}{\eta_{\tau(1)}^{(i)}(a)} \hat{g}_{\tau(1)}^{(i)}(a)$$

$$\leq \frac{2}{\eta_{\tau(k)}^{(i)}(a)}$$

$$= \frac{2}{\eta_{\ell(a,[t_1,t_2])}^{(i)}(a)}$$

Summing over all the actions we obtain the desired inequality. $\qquad \square$

**Lemma 5.8.** *Given a $c > 0$, an $\alpha \in (0, 1)$, a $t \in [\![T]\!]$, and a $\delta > 0$, let $b_t(a) = \frac{c}{n_t(a)^\alpha}$ for all $a \in [\![K]\!]$. Then, with probability at least $1 - \delta$, it holds:*

$$\sum_{\tau=1}^{t} \langle x_\tau, b_\tau \rangle \leq \frac{c}{1-\alpha} K^\alpha t^{1-\alpha} + 4\sqrt{t \log(1/\delta)}.$$

*Proof.* Consider the following inequalities:

$$
\begin{aligned}
\sum_{\tau=1}^{t} b_\tau(a_\tau) &= c \sum_{a \in [\![K]\!]} \sum_{\tau \in [\![t]\!]} \frac{1}{n_\tau(a)^\alpha} \mathbb{I}(a_\tau = a) \\
&= c \sum_{a \in [\![K]\!]} \sum_{k=1}^{n_t(a)} \frac{1}{k^\alpha} \\
&\leq \frac{c}{1-\alpha} \sum_{a \in [\![K]\!]} n_t(a)^{1-\alpha} && \left( \sum_{k=1}^{N} k^{-\alpha} \leq \int_0^N x^{-\alpha} dx \right) \\
&\leq \frac{c}{1-\alpha} K^\alpha t^{1-\alpha} && \text{(Jensen's inequality)}
\end{aligned}
$$

The proof is concluded by using Lemma G.1. $\qquad\square$

# E    Proofs omitted from Section 6

**Theorem 6.1.** *Both in the stochastic and the adversarial setting, with probability at least $1 - 2mT^2\delta_2$ it holds that*

$$V_t \leq 53\sqrt{Kt \log(2/\delta_2)} \quad \forall t \in [\![T]\!].$$

*Proof.* We prove that given an $i \in [\![m]\!]$, it holds:

$$V_t^{(i)} \leq 53\sqrt{Kt \log(2/\delta_2)} \quad \forall t \in [\![T]\!]$$

with probability $1 - 2T^2\delta_2$. Then, a union bound over $i$ completes the proof.

Given an $i \in [\![m]\!]$, we first assume some high-probability events. In particular, we assume that Corollary 5.7 with $\delta = \delta_2$ holds for any interval, and that Lemma 5.8 with $\delta = \delta_2$ holds for all $t \in [\![T]\!]$. This happens with probability at least $1 - 2T^2\delta_2$. We consider two cases. If $V_t^{(i)} \leq 53\sqrt{KT \log(2/\delta_2)}$ for all $t \in [\![T]\!]$, then the statement it is trivially satisfied. Otherwise, there exists an a time $\bar{t}$ for which $V_{\bar{t}}^{(i)} \geq 53\sqrt{Kt \log(1/\delta_2)}$. Clearly, this implies that there exists a $\underline{t} < \bar{t}$ such that $V_t^{(i)} \geq 42\sqrt{Kt \log(2/\delta_2)}$ for all $t \in [\underline{t}, \bar{t}]$ and $V_{\underline{t}-1}^{(i)} \leq 42\sqrt{Kt \log(1/\delta_2)}$. Since $V_t^{(i)} \geq 42\sqrt{Kt \log(1/\delta_2)}$ for all $t \in [\underline{t}, \bar{t}]$ we have that:

$$V_t^{(i)} - 21\sqrt{Kt \log(1/\delta_2)} \geq 42\sqrt{Kt \log(1/\delta_2)} - 21\sqrt{Kt \log(1/\delta_2)} \geq 21\sqrt{Kt \log(1/\delta_2)}$$

and thus $\Gamma_t^{(i)} = 21\sqrt{Kt \log(1/\delta_2)}$ for all $t \in [\underline{t}, \bar{t}]$. Hence, on the interval $t \in [\underline{t}, \bar{t}]$ we known that the learning rate can be lower bounded by a non-increasing function of time as

$$\eta_t^{(i)}(a_t) = \frac{1 + 21\sqrt{Kt \log(1/\delta_2)}}{n_t(a_t)} \geq 21\sqrt{\frac{K \log(1/\delta_2)}{n_t(a_t)}}.$$

This let us use Corollary 5.7 (that we assumed to hold) to show that:

$$V_{[\underline{t},\bar{t}]}^{(i)} \leq \frac{2}{21\sqrt{K\log(1/\delta_2)}} \sum_{a\in[\![K]\!]} \sqrt{n_{\bar{t}}(a)} + \sum_{\tau=\underline{t}}^{\bar{t}} \langle x_\tau, b_\tau \rangle + 4\sqrt{t\log(1/\delta_2)}$$

$$\leq \frac{2\sqrt{K\bar{t}}}{21\sqrt{K\log(1/\delta_2)}} + \sum_{\tau=\underline{t}}^{\bar{t}} \langle x_\tau, b_\tau \rangle + 4\sqrt{t\log(1/\delta_2)} \qquad \text{(Jensen's inequality)}$$

$$\leq \frac{2\sqrt{K\bar{t}}}{21\sqrt{K\log(1/\delta_2)}} + 2\sqrt{2Kt\log(2/\delta_2)} + 8\sqrt{t\log(1/\delta_2)} \qquad \text{(Lemma 5.8)}$$

$$\leq (1/10 + 10)\sqrt{Kt\log(2/\delta_2)}.$$

Now, $V_{\bar{t}}^{(i)} \leq V_{\underline{t}} + V_{[\underline{t},\bar{t}]}^{(i)} \leq (42 + 1/10 + 10)\sqrt{Kt\log(2/\delta)} < 53\sqrt{Kt\log(2/\delta)}$. We thus reached a contradiction and there is no such a $\bar{t}$. The union bound on all $i \in [\![m]\!]$ concludes the proof. $\qquad \square$

**Lemma 6.2.** *In the stochastic setting, with probability at least $1 - 5mKT\delta_2$, it holds that:*

$$|\hat{g}_t^{(i)}(a) - \bar{g}_t^{(i)}(a)| \leq b_t(a) \quad \forall a \in [\![K]\!], t \in [\![T]\!], i \in [\![m]\!]$$

*Proof.* First, we show some concentration inequalities that will be useful in the following. By an Hoeffding's inequality and an union bound with probability at least $1 - mKT\delta_2$, it holds:

$$\left| \frac{1}{n_{t-1}(a)} \sum_{\tau \in \mathcal{T}_{t-1,a}} g_\tau^{(i)}(a) - \bar{g}^{(i)}a \right| \leq \sqrt{\frac{2\log(2/\delta_2)}{n_t(a)}} \quad \forall t \in [\![T]\!], k \in [\![K]\!], i \in [\![m]\!]. \qquad (10)$$

Moreover, by Lemma Lemma G.1 and an union bound, with probability at least $1 - mT\delta_2$, it holds:

$$V_t^{(i)} \leq \sum_{\tau=1}^{t-1} \langle x_\tau, g_\tau^{(i)} \rangle + 4\sqrt{t\log(1/\delta_2)} \quad \forall t \in [\![T]\!], i \in [\![m]\!] \qquad (11)$$

Similarly, by Lemma Lemma G.1 and an union bound, with probability at least $1 - mT\delta_2$, it holds:

$$\sum_{\tau=1}^{t} \langle x_\tau, \bar{g}_\tau^{(i)} \rangle \leq \sum_{\tau=1}^{t} \bar{g}^{(i)}(a_\tau) + 4\sqrt{t\log(1/\delta_2)} \quad \forall t \in [\![T]\!], i \in [\![m]\!] \qquad (12)$$

By Lemma G.2 and an union bound, with probability at least $1 - mT\delta_2$

$$\sum_{\tau=1}^{t} \langle x_\tau, g_\tau^{(i)} \rangle \leq \sum_{\tau=1}^{t} \langle x_\tau, \bar{g}^{(i)} \rangle + 4\sqrt{t\log(1/\delta_2)} \quad \forall t \in [\![T]\!], i \in [\![m]\!] \qquad (13)$$

Finally, by Lemma 5.8 and an union bound, with probability $1 - T\delta_2$, it holds:

$$\sum_{\tau=1}^{t} \langle x_\tau, b_\tau \rangle \leq 2\sqrt{2Kt\log(2/\delta_2)} + 4\sqrt{t\log(1/\delta_2)} \quad \forall t \in [\![T]\!] \qquad (14)$$

In the following, we will assume the the previous events hold, and hence our result holds with probability at least $1 - 5mKT\delta_2$.

First, we show that $V_t^i \leq 21\sqrt{Kt\log(2/\delta_2)}$ for each $t$ and $i$. Our proof works by induction on $t$. Clearly, the inequality holds for $t = 1$. Now, assume that it holds for all $\tau \leq t - 1$. By the definition of $\Gamma_\tau^{(i)}$, the induction assumption implies that $\eta_\tau^{(i)}(a) = \frac{1}{n_\tau(a)}$ for all $a \in [\![K]\!], i \in [\![m]\!]$ and $\tau \leq t - 1$. Then, thanks to Proposition 5.4 we have that:

$$\hat{g}_\tau^{(i)}(a) = \frac{1}{n_{\tau-1}(a)} \sum_{\hat{t}\in\mathcal{T}_{\tau-1,a}} g_{\hat{t}}^{(i)}(a) \quad \forall \tau \leq t - 1. \qquad (15)$$

Hence, by Equation (10), it holds that:

$$\left|\hat{g}_\tau^{(i)}(a) - \bar{g}^{(i)}(a)\right| \leq \sqrt{\frac{2\log(2/\delta_2)}{n_\tau(a)}} \quad \forall \tau \leq t-1.$$

and thus that $\widehat{\mathcal{X}}_\tau^{(i)} \neq \{\emptyset\}$ for all $\tau \leq t-1$. Assuming that the events above holds, consider now the following inequalities:

$$
\begin{aligned}
V_t^{(i)} &= V_{t-1}^{(i)} + g_t^{(i)}(a_t) \\
&\leq \sum_{\tau=1}^{t-1}\langle x_\tau, g_\tau^{(i)}\rangle + g_t^{(i)}(a_t) + 4\sqrt{t\log(1/\delta_2)} && \text{(Equation (11))} \\
&\leq \sum_{\tau=1}^{t-1}\langle x_\tau, g_\tau^{(i)} - \hat{g}_\tau^{(i)}\rangle + \sum_{\tau=1}^{t-1}\langle x_\tau, b_\tau\rangle + g_t^{(i)}(a_t) + 4\sqrt{t\log(1/\delta_2)} && (x_\tau \in \widehat{\mathcal{X}}_\tau^{(i)}) \\
&\leq \sum_{\tau=1}^{t-1}\langle x_\tau, g_\tau^{(i)} - \hat{g}_\tau^{(i)}\rangle + 2\sqrt{2Kt\log(2/\delta_2)} + g_t^{(i)}(a_t) + 8\sqrt{t\log(1/\delta_2)} && \text{(Equation (14))} \\
&\leq \sum_{\tau=1}^{t-1}\langle x_\tau, g_\tau^{(i)} - \hat{g}_\tau^{(i)}\rangle + 2\sqrt{2Kt\log(2/\delta_2)} + 1 + 8\sqrt{t\log(1/\delta_2)} && (g_t^{(i)}(a) \leq 1) \\
&\leq \sum_{\tau=1}^{t-1}\langle x_\tau, \bar{g}^{(i)} - \hat{g}_\tau^{(i)}\rangle + 2\sqrt{2Kt\log(2/\delta_2)} + 1 + 12\sqrt{t\log(1/\delta_2)} && \text{(Equation (13))} \\
&\leq \sum_{\tau=1}^{t-1}(\bar{g}^{(i)}(a_\tau) - \hat{g}_\tau^{(i)}(a_\tau)) + 2\sqrt{2Kt\log(2/\delta_2)} + 1 + 12\sqrt{t\log(1/\delta_2)} && \text{(Equation (12))} \\
&= \sum_{a\in[\![K]\!]}\sum_{\tau=1}^{t-1}(\bar{g}^{(i)}(a) - \hat{g}_\tau^{(i)}(a))\mathbb{I}(a_\tau = a) + 2\sqrt{2Kt\log(2/\delta_2)} + 1 + 12\sqrt{t\log(1/\delta_2)} \\
&\leq \sqrt{2\log(2/\delta_2)}\sum_{a\in[\![K]\!]}\sum_{\tau=1}^{t-1}\frac{1}{\sqrt{n_\tau(a)}}\mathbb{I}(a_\tau = a) + 2\sqrt{2Kt\log(2/\delta_2)} + 1 + 12\sqrt{t\log(1/\delta_2)} \\
&\leq 2\sqrt{2Kt\log(2/\delta_2)} + 2\sqrt{2Kt\log(2/\delta_2)} + 1 + 12\sqrt{t\log(1/\delta_2)}
\end{aligned}
$$

and thus $V_t^{(i)} \leq 21\sqrt{Kt\log(2/\delta_2)}$.

Thus $\Gamma_t^{(i)} = 0$ and $\hat{g}_t^{(i)}(a)$ is the empirical mean of past observations. This concludes the induction step, showing that $V_t^i \leq 21\sqrt{Kt\log(1/\delta_2)}$ for all $t \in [\![T]\!]$, and $\Gamma_t^{(i)} = 0$ for all $t \in [\![T]\!]$ and $i \in [\![m]\!]$.

Now, we proved that with probability $1 - 3mKT\delta_2$, all $\Gamma_t^{(i)} = 0$, and hence by Equation (10) we have that:

$$\left|\hat{g}_t^{(i)}(a) - \bar{g}^{(i)}(a)\right| \leq \sqrt{\frac{2\log(2/\delta_2)}{n_t(a)}} \quad \forall i \in [\![m]\!], t \in [\![T]\!], a \in [\![K]\!]$$

as desired. □

## F  Proofs omitted from Section 7

**Theorem 7.1.** *In the stochastic setting, for any $\epsilon > 0$ Algorithm 1 guarantees that with probability at least $1 - \epsilon$:*

$$R_T \leq 4\sqrt{KT\log(2K/\epsilon)} \quad \text{and} \quad V_t \leq 53\sqrt{Kt\log(28mKT^2/\epsilon)} \quad \forall t \in [\![T]\!].$$

*Proof.* To prove the upper bound on the regret, we simply have to combine Corollary 6.3 with Theorem 4.1. By Corollary 6.3 which probability at least $1 - 5mKT\delta$, it holds $\mathcal{X}^\star \subseteq \cap_{t\in[\![T]\!]}\widehat{\mathcal{X}}_t$.

Moreover, by Theorem 4.1, we have that for each $x \in \mathcal{X}^\star$ with probability at least $1 - \delta_1$:

$$\sum_{t \in [\![T]\!]} \langle f_t, x \rangle - f_t(a_t) \leq 4\sqrt{KT \log(K/\delta_1)}.$$

Let $x^\star = \arg\max_{x \in \mathcal{X}^\star} \sum_{t \in [\![T]\!]} \langle x, f_t \rangle$. Then, by union bound we have that with probability at least $1 - 5mKT\delta_2 - \delta_1$ it holds:

$$\sum_{t \in [\![T]\!]} \langle f_t, x^\star \rangle - f_t(a_t) \leq 4\sqrt{KT \log(K/\delta_1)},$$

proving the bound on the regret. The bound on the violations holds with probability at least $1 - 2mT^2\delta_2$ by Theorem 6.1, and guarantees:

$$V_t \leq 53\sqrt{Kt \log(2/\delta_2)}.$$

By an union bounds on all events, the guarantees hold with probability at least $1 - 7mKT^2\delta_2 - \delta_1$. Thus by taking $\delta_1 = \epsilon/2$ and $\delta_2 = \epsilon/(14mKT^2)$ we obtain the desired result. $\qquad \square$

**Theorem 7.2.** *In the adversarial setting, for any $\epsilon > 0$ Algorithm 1 guarantees that with probability at least $1 - \epsilon$:*

$$\alpha\text{-}R_T \leq 4\sqrt{KT \log(2K/\epsilon)} \quad and \quad V_t \leq 53\sqrt{Kt \log(28mKT^2/\epsilon)} \quad \forall t \in [\![T]\!],$$

*where $\alpha = \rho/(1+\rho)$.*

*Proof.* Combining Theorem 5.2 and Theorem 4.1 readily proves that with probability at least $1 - \delta_1$ we have that for all $\tilde{x} \in \mathcal{X}_\varnothing^\star \subseteq \widehat{\mathcal{X}}_t$, we have:

$$\sum_{t \in [\![T]\!]} \langle f_t, \tilde{x} \rangle - f_t(a_t) \leq 4\sqrt{KT \log(K/\delta_1)}.$$

Let $x^\star = \arg\max_{x \in \Delta_K} \sum_{t \in [\![T]\!]} \langle x, f_t \rangle$. Then, observe that $\bar{x} = \frac{1}{1+\rho}x^\varnothing + \frac{\rho}{1+\rho}x^* \in \mathcal{X}^\varnothing$, where $x^\varnothing(a^\varnothing) = 1$ and $x^\varnothing(a) = 0$ for each $a \neq a^\varnothing$. Then, we have that:

$$\sum_{t \in [\![T]\!]} \langle \bar{x}, f_t \rangle = \sum_{t \in [\![T]\!]} \left\langle \frac{1}{1+\rho}x^\varnothing + \frac{\rho}{1+\rho}x^\star, f_t \right\rangle \geq \frac{\rho}{1+\rho} \sum_{t \in [\![T]\!]} \langle x^\star, f_t \rangle.$$

since $f_t(a^\varnothing) \geq 0$. This proves that with probability at least $1 - \delta_1$:

$$\left(\tfrac{\rho}{1+\rho}\right)\text{-}R_T \leq 4\sqrt{KT \log(K/\delta_1)}.$$

Similarly to the proof of Theorem 7.1, we can prove that the bound on the violations holds with probability at least $1 - 2mT^2\delta_2$ by Theorem 6.1, and give:

$$V_t \leq 53\sqrt{Kt \log(2/\delta_2)}.$$

Overall these events hold with probability at least $1 - 2mT^2\delta_2 - \delta_1$. By defining $\delta_1 = \epsilon/2$ and $\delta_2 = \epsilon/(14mKT^2)$ we have that the desired results hold with probability at least $1 - \epsilon$. $\qquad \square$

**Theorem 7.3.** *Algorithm 1, in the stochastic setting, guarantees that with probability at least $1 - \epsilon$, it holds that:*

$$\mathcal{V}_t^+ \leq 16\sqrt{Kt \log(28mKT^2/\epsilon)} \quad \forall t \in [\![T]\!].$$

*Proof.* Define for each $i \in [\![m]\!]$ and $t \in [\![T]\!]$

$$\mathcal{V}_t^{i,+} := \sum_{\tau=1}^{t} \left[ \langle x_\tau, \bar{g}_\tau^{(i)} \rangle \right]^+.$$

Then, given an $i$ and a $t$ consider the following chain of inequalities:

$$\mathcal{V}_t^{i,+} = \sum_{\tau=1}^t \left[ \langle x_\tau, \bar{g}_\tau^{(i)} \rangle \right]^+$$

$$= \sum_{\tau=1}^t \left[ \langle x_\tau, \bar{g}_\tau^{(i)} - \hat{g}_\tau^{(i)} + \hat{g}_\tau^{(i)} \rangle \right]^+$$

$$= \sum_{\tau=1}^t \left[ \langle x_\tau, \bar{g}_\tau^{(i)} - \hat{g}_\tau^{(i)} \rangle + \langle x_\tau, \hat{g}_\tau^{(i)} \rangle \right]^+$$

$$\leq \sum_{\tau=1}^t \left[ \langle x_\tau, \bar{g}_\tau^{(i)} - \hat{g}_\tau^{(i)} \rangle + \langle x_\tau, b_\tau \rangle \right]^+ \qquad (x_\tau \in \widehat{\mathcal{X}}_\tau)$$

$$\leq \sum_{\tau=1}^t \left[ \langle x_\tau, \bar{g}_\tau^{(i)} - \hat{g}_\tau^{(i)} \rangle \right]^+ + \langle x_\tau, b_\tau \rangle]^+$$

$$\leq 2 \sum_{\tau=1}^t \langle x_\tau, b_\tau \rangle^+ \qquad (\text{Lemma 6.2})$$

where last inequality both hold with probability $1 - 5mKT\delta_2$ jointly for each $i$ and $t$.

Since $b_t = \sqrt{\frac{2 \log(2/\delta_2)}{n_{t-1}(a)}}$ we can apply Lemma 5.8 and an union bound on all $t$ to find that with probability at least $1 - T\delta_2 - 5mKT\delta_2$:

$$\mathcal{V}_t^{i,+} \leq 4\sqrt{2Kt \log(2/\delta_2)} + 8\sqrt{t \log(1/\delta_2)} \quad \forall i \in [\![m]\!], t \in [\![T]\!].$$

Thus, we can conclude that:

$$\mathcal{V}_t^+ \leq 16\sqrt{Kt \log(2/\delta_2)} \quad \forall i \in [\![m]\!], t \in [\![T]\!]$$

with probability at least $1 - 6mKT\delta_2$. Recalling that $\delta_2 = \epsilon/(14mKT^2)$ we obtain the result. $\qquad \square$

## G   Further technical lemmas

**Lemma G.1.** *For any sequence of function $r_t : [\![K]\!] \to [-1, 1]$ which is $t - 1$ predictable and any sequence of randomized strategy $x_t \in \Delta_K$, it holds that with probability at least $1 - \delta$:*

$$\left| \sum_{t \in [\![T]\!]} \langle x_t, r_t \rangle - \sum_{t \in [\![T]\!]} r_t(a_t) \right| \leq 4\sqrt{T \log(1/\delta)}.$$

*Proof.* By definition $\mathbb{E}_{a \sim x_t}[r_t(a)] = \sum_{a \in [\![K]\!]} r_t(a) x_t(a) = \langle x_t, r_t \rangle$. Thus the sequence $X_t = \sum_{\tau=1}^t [r_\tau(a_\tau) - \langle x_\tau, r_\tau \rangle]$ is a martingale and $|X_t - X_{t-1}| \leq 2$. Thus we can apply Azuma inequality and find that with probability at least $1 - \delta$:

$$\left| \sum_{t \in [\![T]\!]} \langle x_t, r_t \rangle - \sum_{t \in [\![T]\!]} r_t(a_t) \right| \leq 4\sqrt{T \log(1/\delta)}.$$

$\qquad \square$

**Lemma G.2.** *For any sequence of randomized strategy $x_t \in \Delta_K$ and any function $\bar{r}(a)$ such that $r_t(a)$ are sampled from a distribution with mean $\bar{r}(a)$, i.e., $\mathbb{E}[r_t(a)] = \bar{r}(a)$ and $\mathbb{P}(|r_t(a)| \leq 1) = 1$, it holds that with probability at least $1 - \delta$:*

$$\left| \sum_{t \in [\![T]\!]} \langle x_t, r_t \rangle - \sum_{t \in [\![T]\!]} \langle x_t, \bar{r} \rangle \right| \leq 4\sqrt{T \log(1/\delta)}.$$

*Proof.* This holds by simple application of Azuma's inequality, similarly to the proof of Lemma G.1.

$\qquad \square$

