# OpenReview forum: "Beyond Primal-Dual Methods in Bandits with Stochastic and Adversarial Constraints"
_NeurIPS.cc/2024/Conference — NeurIPS 2024 poster_

### Official Review · Reviewer_RfbD · 2024-06-21

**Soundness:** 4
**Presentation:** 3
**Contribution:** 2
**Rating:** 4
**Confidence:** 4

**Summary:**

This article tackles the problem of multi-armed bandits with general constraints: at each time step, the learner receives a reward and $m$ costs, corresponding to $m$ independent constraints, and its goal is to maximize the reward while maintaning the cumulative cost corresponding to each constraint at a sub-linear level in the time horizon. To choose the action played at each time step (in the sense of the probability to play each arm), the authors propose an adaptation of Exp-IX, that chooses an action in an optimistic feasible set. Their main contribution consists in proving *best-of-both-worlds* guarantees for this algorithm *without knowing* a slackness parameter, usually assumed as known in the related literature. In this setting, best-of-both world guarantees consist in establishing sub-linear regret with respect to the best feasible probability on one side, and establishing simultaneously a competitive ratio determined by the slackness parameter (which is, again, not known and thus not used by the algorithm) on the other hand, if rewards are adversarial. This is the first result of this kind without knowledge of the Slater’s parameter.

**Strengths:**

I found the paper very clear and well written. The setting and the algorithm are well presented, as well as the results. Clear intuitions are provided, especially in the presentation of the technical results.

**Weaknesses:**

In my opinion, there is a major issue with the results,which is that they are obtained with a modified definition of the Slater’s parameter. As stated by the authors, in the literature it is generally defined with the set of probabilities allocated to each arm, while in the paper it is defined with the arm set. While the author claim that this definition is « slightly stronger », I disagree with this claim. It seems to me that on the contrary the problem becomes much simpler if we know in advance a set of K atoms in the simplex that contain for sure a safe policy (even the safest in a sense). To me, this seems like trading a strong assumption (knowing the Slater’s parameter) by another strong assumption. This assumptions also considerably restricts the class of problems that can be considered. Furthermore, from my current understanding of the arguments it seems that the assumption is not just an artifact of the analysis and that the results would not be guaranteed to work if the infimum in the Slater’s condition was achieved by a mixed strategy.

In itself, I find the paper interesting despite this limitation, but I believe that this should be discussed more thouroughly because it reduces the potential impact of the paper.

**Questions:**

* Please elaborate on the definition of the Slater’s condition. In particular, I am interested in understanding if the definition is indeed necessary for the guarantees to hold of if it is an artifact of the analysis.

* You claim that the competitive ratio matches the lower bound obtained in another paper. However, this seems inexact to me because the lower bound do not consider the same definition of the slackness parameter. I am missing something?

---

> ### Author Rebuttal · Authors · 2024-08-07
>
> We thank the reviewer for their comments about the paper.
>
> We agree that a more detailed discussion on this point is necessary and will include an extended version of our response in an additional paragraph. However, we would like to emphasize that the primary contribution of our paper is demonstrating that, perhaps surprisingly, a new and more natural **UCB-style approach** can be effectively employed in the **adversarial** BwK setup.
>
> ### On Slater’s condition:
>
> First, we disagree with the claim that our definition of Slater’s condition “*greatly simplifies the problem*”. While, as the reviewer noted, this may facilitate the identification of a safe strategy, this aspect is not the most challenging part of the problem. Indeed, this can be easily done by simply estimating the cost of all the arms. For example, in the BwK framework, the learner knows a safe action that satisfies all the constraints at the same time, and yet many challenges remain to be addressed.
>
> On a minor note, we point out that in the stochastic setting, Slater’s condition is not even necessary and only pertains to the adversarial setting.
>
> Moreover, we would like to highlight that our definition of Slater’s condition coincides with the one on mixed strategies in the case of a single cost/constraint.
>
> Finally, we would like to remark that in most works (e.g., in the BwK literature), it is assumed that there exists a void action that does not use resources, and hence satisfies all the constraints. This assumption is consistent with our definition.
>
> ### On the lower bound:
>
> Regarding the reviewer’s second question, it is true that the paper from which we borrow the lower bound uses a different definition of Slater's parameter. However, in the instance used in the lower bound, the strategy maximizing the feasibility margin puts all the probability on a single action. We will add a remark on this point to avoid confusion. This supports our assertion that our definition of Slater’s parameter is not significantly weaker than the one commonly used in the literature.

---

> > ### Comment · Reviewer_RfbD · 2024-08-12
> >
> > Thank you very much for your response, I have no more question for now.

---

> > > ### Author Response · Authors · 2024-08-14
> > >
> > > Thank you for the time dedicated to our paper. We think we have addressed all the weaknesses highlighted by the reviewer. Based on our responses, we respectfully ask the reviewer to consider adjusting their evaluation. Thank you.

---

### Official Review · Reviewer_EbNg · 2024-07-12

**Soundness:** 3
**Presentation:** 3
**Contribution:** 2
**Rating:** 5
**Confidence:** 4

**Summary:**

This paper studies the problem of multi-armed bandits under long-term constraints.
They give an algorithm with regret and violation guarantees simultaneously for both stochastic and adversarial constraints (i.e. best-of-both-worlds guarantees).
Specifically, they guarantee $\tilde{O}(\sqrt{T})$ regret and violation under stochastic constraints (without Slater's condition), as well as $\tilde{O}(\sqrt{T})$ $\alpha$-regret and violation under adversarial constraints (with Slater's condition with parameter $\rho$ where $\alpha=\frac{\rho}{\rho +1}$).

**Strengths:**

- The paper appears to improve on [11] in that it's regret scales logarithmic with the number of constraints and doesn't require Slater's condition for stochastic constraints.
- The exposition is clear and there are nice insights from the theoretical results.

**Weaknesses:**

My main concern is that the paper is missing key literature that render the contributions rather limited.
More details in the following:
- [R1] gave a similar algorithm that uses an "optimistic set" to overestimate the constraint and then playing regret minimizer within this set. They showed $\tilde{O}(\sqrt{T})$ regret and violation without Slater's condition. In light of this, it appears the only contribution is extending [R1] to adversarial costs (and more restrictive finite action set). Although useful, this result is more limited than claimed.
- The line of literature on stochastic bandits with round-wise constraints (e.g. [R2],[R3],[R4]) is directly relavant because it considers stochastic constraints with stronger constraint satisfaction guarantees (i.e. no violation).
- The abstract claims that their particular problem setting generalizes Bandits with Knapsacks, which is unclear to me. Without this being the case, it seems that the only improvement on the literature would be for the recent arXiv paper [11].

[R1] Gangrade et al, "Safe Linear Bandits over Unknown Polytopes", COLT 2024 (first appeared on arXiv in 2022 as 2209.13694).

[R2] Amani et al, "Linear Stochastic Bandits Under Safety Constraints", NeurIPS 2019.

[R3] Pacchiano et al, "Stochastic Bandits with Linear Constraints", AISTATS 2021.

[R4] Moradipari et al, "Safe Linear Thompson Sampling With Side Information", IEEE TSP 2021.

**Questions:**

- Could you provide more clarification on how this setting generalizes Bandits with Knapsacks?
- Given the references I gave, could you provide any insight on how this work contributes to this literature?

**Limitations:**

I did not see clear discussion of limitations. See the "Weaknesses" section for my discussion on limitations.

---

> ### Author Rebuttal · Authors · 2024-08-07
>
> Thank you for the positive feedback on our work.
>
> ### On related works:
>
> We will gladly include the related works highlighted by the reviewer in the final version of the paper. However, we do not consider these works to be technically very related to ours. We outline the main reasons below.
>
> First, we disagree with the claim that “*it appears the only contribution is extending [R1] to adversarial costs (and more restrictive finite action set). Although useful, this result is more limited than claimed.*” Indeed, it is well-known since the seminal work of Mannor et al. [1] that the adversarial setting is far more challenging than the stochastic one. We design a minimax optimal algorithm that provides optimal rates under **stochastic and adversarial** constraints. This was explicitly indicated as one **key question for the BwK set-up** in the Immorlica et al. paper [4].
>
> The papers pointed out by the reviewer only consider the stochastic setting, and the techniques that they propose clearly fail to carry over to the adversarial case, thereby rendering it impossible to give guarantees for BOTH stochastic & adversarial inputs. Extending the techniques of [R1] to the adversarial case is a non-trivial task, and it is unclear whether it is even feasible. In our paper, we use entirely different techniques. Therefore, we strongly disagree with characterizing our results as “*extending the results of [R1] to adversarial costs.*”
>
> Even if we just look at the stochastic setting, there are multiple points in which our work differs from the set of papers suggested by the reviewer. Some notable examples are the following:
> * [R2] provide $O(T^{2/3})$ regret guarantees in the case of unknown $\Delta$ ($\Delta$ in [R2] takes the role of our Slater’s parameter), while we always achieve $O(\sqrt{T})$ regret.
> * [R3] works with stage wise constraints and thus has to assume the knowledge of a safe action.
>
> Moreover, the stage wise guarantees are not stronger than ours, since they are defined on the expected costs (expectation with respect to the environment and the inner randomization of the algorithm), while we define violations on the realizations, and thus it is clearly impossible to achieve no violations at all rounds.
>
> ### On the relationship with BwK:
>
> We omitted a formal discussion on this matter due to space constraints, but we will include it in the final version of the paper using the extra space. We extend the BwK framework along two key directions:
> 1. We do not assume knowledge of Slater’s parameter. In the classical BwK framework this is assumed to be known. Indeed, assuming that you know the budget $B$ and the time horizon $T$, the per budget constraint is $cost_t \le B/T:=\rho$, and there is a void action which always provides a cost of zero.
> 2. We can handle both positive and negative costs (some prior literature refers to this problem as bandits with knapsacks with non-monotonic resource utilization; see [2, 3]).
>
> Finally, we observe that we can implement hard constraints (i.e., no violation) by virtually augmenting the costs by $O(1/\sqrt{T})$.
>
> ### References
>
> [1] Mannor, Shie, John N. Tsitsiklis, and Jia Yuan Yu. "Online Learning with Sample Path Constraints." Journal of Machine Learning Research 10.3 (2009).
>
> [2] Kumar, Raunak, and Robert Kleinberg. "Non-monotonic resource utilization in the bandits with knapsacks problem." Advances in Neural Information Processing Systems 35 (2022): 19248-19259.
>
> [3] Bernasconi, M., Castiglioni, M., Celli, A., & Fusco, F. (2022). “Bandits with replenishable knapsacks: the best of both worlds.” The Twelfth International Conference on Learning Representations.
>
> [4] Immorlica, N., Sankararaman, K., Schapire, R., & Slivkins, A. (2022). “Adversarial bandits with knapsacks”. Journal of the ACM, 69(6), 1-47.

---

> ### Comment · Reviewer_EbNg · 2024-08-07
> **Thank you for your response; concerns about contribution**
>
> Thank you for the response.
>
> To be clear, my main concern is in the contributions of the proposed algorithm with respect to [R1], which uses a similar algorithm for the case of stochastic costs and stochastic constraints. Indeed, both algorithms "overestimate" the constraint set and then run a regret minimizer within this overestimated set. The main difference is that [R1] uses a UCB based regret minimizer (to handle the stochastic rewards), while this paper uses an adversarial bandit regret minimizer (to handle adversarial rewards). Furthermore, *[R1] gives similar regret and violation bounds ($\tilde{O}(\sqrt{T})$ regret and violation) and also doesn't require Slater's condition.* One of the main claims is that this paper gives the first algorithm with $\tilde{O}(\sqrt{T})$ regret without Slater's condition in stochastic settings, which appears to be incorrect.
>
> As for my question about the relationship with BwK, I was more interested in how exactly this setting generalizes BwK as it wasn't immediately clear to me. I think I cleared this up by looking at prior works so this is not an issue, other than that I believe it needs to be stated more clearly.

---

> > ### Author Response · Authors · 2024-08-08
> >
> > We thank the reviewer for the timely answer. We think the reviewer is referring to our sentence (line 71): “Moreover, in stochastic settings, it is the first algorithm to provide $\tilde O(\sqrt{T})$ regret without requiring Slater’s condition.” We acknowledge that the original statement can be misleading when taken out of context. What we intended to convey is that our algorithm is the first **best-of-both-worlds** approach that, in the stochastic case, does not require the Slater condition while still achieving $\tilde O(\sqrt{T})$ regret and constraint violations. We agree that this should have been communicated more clearly, and we are grateful to the reviewer for bringing this to our attention.
> >
> > However we do not agree that our main claim is that “ this paper gives the first algorithm with regret without Slater's condition in stochastic settings”. This is not one of our primary contributions at all! Our main contribution is designing a simple and intuitive algorithm that provides best-of-both-worlds guarantees in bandits with general constraints while also being minimax optimal. This algorithm is UCB-like in the stochastic constraints case, but the reviewer's intuition that we can simply swap a UCB-like regret minimizer for stochastic rewards with an adversarial one (since we need to handle adversarial inputs) is wrong. Our algorithm indeed needs to account for the adversarial nature of the constraints, and it does so by automatically switching to alternative methods of estimating them, adjusting the computation of estimates and increasing the learning rate as needed. The techniques we developed to prove that this approach works and achieves optimal rates in adversarial and stochastic settings are far from trivial and do not rely on the techniques used in [R1]. We hope that a closer examination of our paper will convince the reviewer of this point.
> >
> > We therefore propose to revise the claim in line 71 and include a discussion of the related works for the stochastic setting, as suggested by the reviewer, including but not limited to [R1] (we initially overlooked [R1] since it was only recently published at COLT 2024, but we agree that it relevant to the discussion).

---

> > > ### Comment · Reviewer_EbNg · 2024-08-12
> > >
> > > Thank you for the clarifying response. I do believe that the paper makes useful contribution to the literature on best-of-both worlds guarantees in bandits with long term constraint. I am raising my score under the expectation that [R1] and [R6] are cited in reference to the conceptual similarities in algorithm design (i.e. use of "optimistic estimations" of the constraints) and the relevant results for stochastic settings. (I'm also including [R6], which was pointed out by reviewer Q9L2, as it also uses this design approach and was published back in 2022.)
> > >
> > > [R6] Chen et al., Strategies for Safe Multi-Armed Bandits with Logarithmic Regret and Risk, ICML 22

---

### Official Review · Reviewer_EcmS · 2024-07-27

**Soundness:** 4
**Presentation:** 3
**Contribution:** 3
**Rating:** 7
**Confidence:** 3

**Summary:**

This work studies the bandit with constraints problem where the authors consider two possible settings for the constraints -- one where the constraints are stochastic, sampled i.i.d. from some unknown distribution, and one where the constraints are adversarial. The rewards are always assumed to be generated by an oblivious adversary. The authors present a novel algorithm which is able to achieve the optimal competitive ratio, up to O(\sqrt{T}) regret in the constraints and objective, in the adversarial constraints setting and O(\sqrt{T}) regret in the constraints and objective in the stochastic setting which is also min-max optimal.

**Strengths:**

To the best of my knowledge the proposed algorithms and regret bounds are novel. The results are nearly min-max optimal (up to small logarithmic factors in K). The presentation of the work is overall good and there seems to be good technical novelty in developing the algorithms. The challenge in this work for achieving best of both worlds regret for the constraints is in constructing the right empirical estimators for the constraints. This is explained well in Section 6. In particular there is tension between tightly bounding V_t and having an estimator which constraints well in the stochastic setting. The main novelty seems to be the choice of adaptive step size in the definition of the empirical constraints which leads to the neat proof of Lemma 6.2 and Theorem 6.1.

**Weaknesses:**

I do not find any major weaknesses in this work. Perhaps the authors can do slightly better in highlighting the tension between bounding V_t and concentration of the empirical estimators. It was also not entirely clear to me what is the benefit of viewing the estimator update as a variant of gradient descent.

Minor typos:
line 517: This let us...
line 517: The underscored t subscript is larger
Equation 10: \bar g^{i}a -- brackets are missing

**Questions:**

Is it possible to show some type of gap-dependent regret guarantees for the regret of the constraint violation using the same type of estimator? Intuitively it seems like there would still be fast enough concentration of the empirical estimator of the constraints, so that infeasible actions are eliminated quickly in case of a large gap, however, it's not completely obvious how to handle some parts of bound for V_t.

**Limitations:**

Limitations are appropriately addressed and there is no obvious negative societal impact as this work is theoretical.

---

> ### Author Rebuttal · Authors · 2024-08-07
>
> Thanks for the positive comments about our paper.
>
> **On the OGD interpretation of the estimator updates:** the main benefit of viewing the update as a variant of gradient descent is that it allows us to simplify the analysis in the proofs. We will add a remark in the main paper to clarify this point.
>
> **On the instance-dependent result in the stochastic case:** Note that it is not trivial to find a good definition of “gap” in settings with long-term constraints. The presence of constraints implies that the optimal strategy may be a mixed distribution, and thus, a  “second best” arm may not exist. There are some works that try to conjugate a logarithmic dependence in $T$ with some instance-dependent quantities (see e.g. [1,2]). However, these works require strong assumptions to establish such dependencies, making this direction less explored compared to the standard MAB setting. We will add some comments about this in the final version of the paper.
>
>
>
> [1] Sankararaman, K. A., & Slivkins, A. (2021). “Bandits with knapsacks beyond the worst case.” Advances in Neural Information Processing Systems, 34, 23191-23204.
>
> [2] Li, Xiaocheng, Chunlin Sun, and Yinyu Ye. “The symmetry between arms and knapsacks: A primal-dual approach for bandits with knapsacks.” International Conference on Machine Learning. PMLR, 2021.

---

### Official Review · Reviewer_Q9L2 · 2024-07-29

**Soundness:** 4
**Presentation:** 3
**Contribution:** 3
**Rating:** 7
**Confidence:** 4

**Summary:**

This paper studies multi-armed bandits with cumulative cost connstraints, with the focus of designing a 'best of both worlds' algorithm that has small constraint violations in both the stochastic and adverasarial settings (where the constraints as well as rewards can vary with time arbitrarily), and attain either low-regret for the stochastic setting, or low $alpha$-regret for the adversarial setting (i.e., ensure that the reward accrued is at least $\alpha$ times the best-in-hindsight reward accrued by a constant unconstrained assignment probability x, up to $o(T)$ terms). The main desiderata is to design methods that avoid explicit knowledge of a slater parameter, and avoid polynomial dependence on m, the number of unknown constraints, in the regret and violations, both of which are cons of prior work using the  Langrangian BwK design.

Throughout, the setup adopted is linearised, i.e., at each time, the authors select a distribution $x_t$ over the actions [1:K], and the action $a_t$ is selected by sampling from $x_t$. A bandit feedback of both the reward, $f_t(a)$, and the constaint violations $g_{t}^i(a), i \in [1:m]$ is assumed, where these functions are stochastic perturbations of an expected function in the stochastic setting, and are selected arbitrarily in the adversarial setting. Throughout, it is assumed that $f_t \in [0,1]$ and $g\_t^i \in [-1,1]$. The constraint is of the form $\forall i, g_t^i \le 0$, Net violations are defined as $V_T = \max_{i} V_T^{i}, $ where $V_T^i = \sum_t g_t^i(a_t).$ Note that this equals $\sum_t \langle x_t, g_t^i\rangle,$ up to a $\sqrt{T}$ concentration term. In the adverarial case, the paper studies the regret $\frac{\rho}{1+\rho} \max_{x} \sum f_t(x) - \sum f_t(a_t),$ where $\rho$ is the minimax Slater parameter $\rho:=  \max_a \min_{i,t} (-g_{t}^i(a))$.

The overall strategy adopted by the paper is very natural: for each time, the method constructs a set of "plausibly" feasible distributions as $\widehat{\mathcal{X}}\_t = \{x \in \Delta\_K : \forall i, \langle x, \hat{g}\_t^i - b\_t\rangle \le 0\}$, where $\hat{g}\_t^i$ serve as "estimates" for $g\_t^i,$ and $b$ is a nonnegative 'bonus' that ensures optimism in the stochastic scenario. The method then just selects actions from $\widehat{\mathcal{X}}\_t$, using a (small modification of) the EXP-IX strategy, thus ensuring low regret relative to any constant $x \in \bigcap \widehat{\mathcal{X}}\_t$. The main results then need to select $\hat{g}_t^i$ and $b_t$ in such a way that

1. In the stochastic case, $\mathcal{X}^* = \{x : \forall i, \langle \mathbb{E}[g_t^i], x\rangle \le 0\}$ remains within this intersection of $\widehat{\mathcal{X}}s$, and
2. In the adverasrial case, $\frac{\delta\_{a^{\emptyset}}}{1+\rho} + \frac{\rho}{1+\rho} \mathcal{X}$ remains within the same.
3. The net violation is small.

As is natural, the $b_t(a)$ is set to just $\sqrt{ \log(mT^2/\delta)/n_{t}(a)},$ where $n_t(a)$ is the number of times action $a$ has been played up to time $t$. The $\hat{g}\_t^i$ are estimated using OGD as $$ \hat{g}\_{t+1}^i(a)  =  \hat{g}\_t^i(a)\mathbf{1}\{a\_t \neq a\} + ( (1-\eta\_{t}^i(a) \hat{g}\_t^i(a) + \eta\_t^i(a) g\_t^i(a))\mathbf{1}\{a\_t = a\}.$$ The chief question becomes how to select the learning rates $\eta\_t^i(a)$. The basic tension is illustrated thus: in the stochastic case, we'd like $\eta\_t^i$ to be roughly $1/n\_{t}(a),$ so that $\hat{g}\_t^i(a)$ is essentially the empirical average of the feedback. However, in the adversarial case, such a rate would be way too slow, and one needs quicker adaptation. The authors elegantly resolve this dilemma by setting $ \eta\_t^{i}(a) =(1+\Gamma\_t^i)/n\_t^i,$ where $$ \Gamma\_t^t = \min( \zeta\_t , \max( 0, V\_t^i - \zeta\_t )), \textrm{ where } \zeta\_t = 21\sqrt{Kt \log(mT^2/\delta) }$$

For the stochastic case, the analysis directly shows that, whp, $V\_t^i$ never crosses the $21\sqrt{Kt}$ threshold, and thus the desired $\eta\_t = 1/n\_t(a)$ is retained. For the adversarial case, the analysis proceeds by first controlling the violation over any interval in terms of the updates to $\hat{g}\_t$ over the same, and then arguing via contradition by showing that that if $V_\tau/\sqrt{K\tau}$ is ever too large, then $\eta_t$ is large over a significantly sized interval before it, which in turn implies strong adaptation and a small $V_\tau$. Finally, the point 1 follows by the choice of $b_t(a) \sim 1/\sqrt{n\_t(a)}$, and 2 follows directly from the boundedness of the $g\_t^i$, by interpreting $\hat{g}_t^i$ as a weighted sum of the observed $g\_t^i(a_t)$s with net weight $1$.

**Strengths:**

The subject of the paper is of course very peritnent to the online learning subcommunity at Neurips. I think the technical contribution of the paper, i.e., the design of a BOBW method for the bandits with cumulative constraints that avoids knowledge of Slater parameters is quite interesting. Furthermore, the removal of the poly(m) factors from the ensuing bounds is an important improvement in the same.
explained
I find the paper to be quite well written. The thought process behind the design of the method is well explained, and the intuition behind the design of the learning rate is clear. Prior work is contextualised well, and the contribution of the paper is clearly outlined. To my understanding, the proofs are correct.

The design of the scheme, and the analysis, are simple, but well motivated and elegant. Altogether, while I don't see this paper as especially groundbreaking, I do find it to be a well executed contribution that develops a simple idea and does interesting things with it to clean up the theory of bandits with cumulative constraints.

**Weaknesses:**

Perhaps the main weakness is that the structure of $\Gamma\_t^i$ bakes in the large constant 21 (and this of course leads to the large constant 53 in the bound). I find it natural to wonder if this constant is necessary, or if can be set in a path-dependent way that avoids both the large constant, and potentially offers improvements if the $g$s are somehow 'nice'. That said, I think that such a lacuna is not a major weakness, and such questions are find being left to follow-up work.

**Questions:**

Comment: Comment: the roundiwse violation bound on V_T^+ of theorem 7.3 has previously been observed for the stochastic setup in the "safe bandit" literature, and this point of contact should perhaps be acknoweldged. See, e.g., [1,2]. The method these papers study is Agarwal and Devanur's style of "doubly-optmistic" selection.

[1]: Chen et al., Strategies for Safe Multi-Armed Bandits with Logarithmic Regret and Risk, ICML 22
[2]: Gangrade et al., Safe Linear Bandits over Unknown Polytopes, COLT 24

**Limitations:**

This is fine.

---

> ### Author Rebuttal · Authors · 2024-08-07
>
> Thanks for the positive comments about our work and for pointing out the line of research on safe bandits. We will acknowledge it in the final version of the paper.
>
> On the constants: we didn’t try to optimize the constants as we were mainly interested in achieving asymptotically optimal regret rates. We agree that it would be interesting to explore the methodology suggested by the Reviewer; we will add it as a future research direction.

---

### Decision · Program_Chairs · 2024-09-25

**Decision:**

Accept (poster)

**Comment:**

This paper makes a clear contribution to bandits with cumulative constraints. It overcomes prior difficulties in obtaining BOBW results by using the existence of a safe arm (instead of a safe mixture).
The literature is not settled on a standard assumption and I ask the authors to make it very clear in the main body of the final version that this is a stronger assumption than in some of the prior work they compare themselves to.

Nonetheless, it is an assumption that is not uncommon in the literature and achieving BOBW in this regime is a novel and significant contribution. The reviewers are in agreement that the paper is well written and that the results are significant.